# Relative Survival, Conditional Survival, and Causes of Death in Patients with Early Gastric Cancer, with a Focus on Differences Between Cardia and Non-Cardia Cancer

**DOI:** 10.3390/cancers16244262

**Published:** 2024-12-21

**Authors:** Anas Elgenidy, Omar Alomari, Mohamed Marey Hesn, Anas Khaled, Sarah A. Nada, Mostafa Elsayed, Ali Mahmoud, Mohammed Al-mahdi Al-kurdi, Ahmed M. Afifi, George Cholankeril

**Affiliations:** 1Department of Neurology, Cairo University, Cairo 11652, Egypt; anas.elgenidy@gmail.com; 2Hamidiye International School of Medicine, University of Health Sciences, 34668 Istanbul, Turkey; dromari2001@gmail.com; 3Damietta Faculty of Medicine, Al-Azhar University, Damietta 34517, Egypt; mohammedhesn580@gmail.com (M.M.H.); anskhaled31@gmail.com (A.K.); 4Faculty of Medicine, Menofia University, Shebin El-Kom 32861, Egypt; sarah.nada.6318@gmail.com; 5Faculty of Medicine, Zagazig University, Zagazig 44691, Egypt; mos.mahmoud97@gmail.com; 6College of Medicine and Life Sciences, University of Toledo, Toledo, OH 43614, USA; ali.mahmoud2@rockets.utoledo.edu; 7Faculty of Medicine, University of Aleppo, Aleppo 15310, Syria; mohammedmahdikurdi@gmail.com; 8University of Toledo Medical Center, Toledo, OH 43614, USA; 9Section of Gastroenterology and Hepatology, Margaret M. and Albert B. Alkek Department of Medicine, Baylor College of Medicine, Houston, TX 77030, USA; george.cholankeril@bcm.edu

**Keywords:** causes of death, epidemiology, mortality, Gastric cancer, SEER

## Abstract

Gastric cancer is a significant global health concern, with over one million new cases and approximately 769,000 deaths in 2020. This study is the first to thoroughly investigate the causes of death in early-stage gastric cancer, specifically comparing cancers located at the cardia and non-cardia regions of the stomach. By analyzing data from 9721 patients, we aim to understand the different survival outcomes and causes of death between these two groups. Our findings highlight important survival patterns and emphasize the necessity of early diagnosis and tailored treatment plans. This research provides crucial insights that can help healthcare professionals improve the management and prognosis of gastric cancer patients.

## 1. Introduction

Gastric cancer (GC) is one of the most common malignancies and ranks as the third leading cause of cancer-related deaths globally [1,2,3,4]. About 26,500 additional instances of GC (15,930 in men and 10,570 in women) are anticipated in the United States in 2023 [5]. GC caused approximately 11,130 deaths (6690 males and 4440 women) in 2023 [5]. While less common in the US, GC remains prevalent in East Asia [5]. Early diagnosis and surgical intervention are crucial for improving survival rates, yet GC is often detected at advanced stages.

Gastric cancer can be classified into two primary types: proximal gastric (cardia) cancer (CGC) and distal gastric (non-cardia) cancer (NCGC) [6]. These categories differ in incidence rates, risk factors, and clinical characteristics. CGC is more common in developed countries and is associated with obesity and gastroesophageal reflux disease, whereas NCGC is more prevalent in underdeveloped regions and is strongly linked to Helicobacter pylori infection and dietary factors such as the consumption of salty and smoked foods [1,6,7,8,9,10,11,12].

Based on data extracted from the Surveillance, Epidemiology, and End Results (SEER) program, the incidence rates of GC in the United States have exhibited a consistent decline from 1992 to 2019, with an annual reduction of 1.23% [10]. This decline was chiefly attributed to a decrease in NCGC, which experienced an annual decline of 1.5%, as opposed to CGC, which exhibited an annual decline of 0.5% [10]. Notably, a rise in GC incidence rates has been observed in age groups below 50, with the most substantial increase manifesting within the under-40 age group. Remarkably, NCGC accounts for 74.3% of GC cases among individuals under 50, with the incidence rate rising in correlation with age [10].

Recently, Lv et al. (2021) reported that cancer-specific mortality was higher in the cardia cancer group compared to the non-cardia cancer group and also noted the poor prognosis of early cardia cancer, which is associated with less differentiation and worse pathologic types [1].

Many studies, which examined the molecular characteristics of cardia cancer, showed increased expression of HER2, Sirt1, and TP53 genes and significantly amplified PAKI and KRAS genes, which indicates poor prognosis [13,14]. Other studies reported increased infiltration of cardia cancer compared to non-cardia cancer [15,16].

Despite being a critical determinant for successful management, the early diagnosis rate for GC remains low, hindering our understanding of early GC specifically. Early diagnosis and subsequent surgical resection are paramount in the effective management of GC, emphasizing the need for targeted investigations into the early stages of the disease. However, the scarcity of studies addressing early GC reflects the difficulties in achieving timely diagnoses, further underscoring the urgency and importance of new research in bridging this gap.

Also, survival rates are key indicators of the effectiveness of treatment strategies. Examining the survival rates of both CGC and NCGC patients will provide a valuable insight into the disease’s trajectory. Beyond survival rates, understanding the fundamental differences between cardia and non-cardia gastric cancer is paramount. From etiological factors to clinical manifestations, recognizing these distinctions is pivotal for personalized and targeted approaches to patient care.

The primary aim of this study is to investigate the differences between cardia and non-cardia gastric cancer, focusing on causes of death among patients, survival rates, conditional survival, and fundamental distinctions between these subtypes with a specific emphasis on early gastric cancer (stage I) based on the AJCC 6th TNM staging [1]. Our research focuses on early gastric cardia cancer due to a notable gap in the existing literature regarding the limited number of studies and the challenges associated with early diagnosis in GC as a whole.

Utilizing the extensive dataset from the SEER program, our research seeks to provide a nuanced understanding of demographic and tumor-related variables, contributing essential insights that can inform tailored therapeutic approaches and improve overall outcomes for individuals diagnosed with gastric cancer.

## 2. Materials and Methods

In our retrospective investigation, we employed the SEER*stat software (National Cancer Institute (Bethesda, MD, USA)), specifically version 8.3.92. This software facilitated our access to the SEER 17 plus dataset, specifically, the Nov 2019 sub-dataset spanning the years 2000 to 2019. The flow of the study is summarized on Figure 1. The study data was procured from the publicly accessible SEER 17 registry, administered by the National Cancer Institute (NCI). In view of the nature of our study and the utilization of anonymized data, the approval of the institutional review board was deemed unnecessary. The cumulative count of instances identified as GC encompassed a total of 49,276 cases spanning the years 2000 to 2019. Subsequently, following confirmation by the International Classification of Disease (ICD), a subset of 34,212 cases was classified as adenocarcinoma, including the following types: 8140/3 (Adenocarcinoma, NOS), 8144/3 (Adenocarcinoma, intestinal type), 8145/3 (Carcinoma, diffuse type), 8211/3 (Tubular adenocarcinoma), 8260/3 (Papillary adenocarcinoma, NOS), 8480/3 (Mucinous adenocarcinoma), and 8490/3 (Signet ring cell carcinoma) [1]. The treatment approach for gastric cancer primarily relies on the TNM staging system. This study encompassed patients falling within Stage 1 as per the AJCC third edition (2000–2003), sixth edition (2004–2015), seventh edition (2016–2017), and eighth edition (2018 onwards). Specifically focusing on early GC, these patients were categorized into four distinct groups: CGC, NCGC, not otherwise specified GC, and GC with overlapping lesions. Notably, the classification of NCGC is anatomical, and it includes subcategories such as the fundus of stomach (C16.1), body of stomach (C16.2), gastric antrum (C16.3), and pylorus (C16.4).

These are the parts of the stomach located below the cardia, which is a 3 cm transition zone between the esophagus and stomach. Excluded from the analysis were groups comprising not otherwise specified GC and GC with overlapping lesions. The final study cohort consisted of 9721 patients diagnosed with early GC, with 4384 individuals exhibiting CGC and 5337 individuals presenting NCGC.

### 2.1. Inclusion Criteria

The inclusion criteria comprised patients diagnosed with GC between 2000 and 2019, displaying a well-defined tumor site, gastric adenocarcinoma, early-stage cancer, primary cardia or non-cardia location, and a survival period exceeding 1 month.

### 2.2. Statistical Analysis

The standardized mortality ratios (SMR) and corresponding 95% confidence intervals (CI) were computed using SEER*stat software version 8.3.9.2. The SMR (O/E) ratio compares the observed deaths (O) among patients diagnosed with CGC or NCGC to the expected deaths (E) in a demographically similar group, adjusted for age, over the same time period.

Relative survival is a key measure of cancer survival that excludes other mortality factors. It is calculated as the ratio of observed survival in cancer patients to the expected survival in a comparable group of cancer-free individuals [17,18,19,20]. This calculation assumes independent and competing causes of death. By considering overall survival rates within the target population based on factors such as sex, age, race, and the date age was recorded, relative survival provides a normalized view of cancer outcomes. Conditional survival, on the other hand, refers to the probability of surviving a specific time frame (e.g., 5 years) after already surviving a set duration post-cancer diagnosis (e.g., 1, 3, or 5 years) [21]. Where applicable, conditional survival in the analysis was derived from the concept of relative survival [18].

Observed and relative survival rates at 1 and 5 years’ post-diagnosis for both cardia and non-cardia cancers were calculated, with further categorization by age and sex. Additionally, a 3-year conditional survival analysis was conducted for individuals who survived 1 and 5 years after their initial diagnosis.

All statistical evaluations were conducted using a two-sided approach, and statistical significance was determined by a *p* value below 0.05 (*p* < 0.05). Significantly elevated risk was established when observed deaths resulting from a specific cause subsequent to CGC or NCGC diagnosis surpassed the anticipated deaths of the same cause within the general population, with a *p* value less than 0.05 (*p* < 0.05).

### 2.3. Data Availability

The study’s data was sourced from the SEER program, accessible at www.seer.cancer.gov.

### 2.4. Ethical Approval

The study exclusively employed data from the SEER database, and as such, no human experiments were conducted. Consequently, the study did not necessitate ethical approval or the procurement of informed consent.

## 3. Results

### 3.1. Baseline Characteristics

A comprehensive cohort of 9721 patients diagnosed with GC was encompassed within this study. Among these, 4384 patients (45.1%) exhibited a diagnosis of CGC, whereas 5337 patients (54.9%) manifested NCGC presentations. The majority of the enrolled patients were aged above 65 years old, comprising 2810 (64.1%) in the cardia group and 3748 (70.2%) in the non-cardia group. Additionally, a substantial proportion identified as white, accounting for 3886 (88.6%) and 3044 (57%) in the cardia and non-cardia groups, respectively. A significant number of patients were married, with 2696 (61.5%) in the cardia group and 2885 (54.1%) in the non-cardia group. Moreover, in terms of gender distribution, 3419 (78%) were males in the cardia group, while 2889 (54.1%) were males in the non-cardia group. A comprehensive overview of demographic characteristics for both cardia and non-cardia patients is presented in Table 1.

During the follow-up period, a total of 5381 individuals (55.4%) died due to various causes. The most pronounced surge in mortality for both groups transpired within the initial year subsequent to diagnosis (Figure 2). Among these cases, 48.9% pertained to CGC with an average age of death recorded at 74.37 years, while the remaining 51.1% corresponded to NCGC patients, characterized by an average age of death at 78.75 years.

In the case of CGC, the most deaths (1063; 40.4%) happened in the first year after the diagnosis of GC. Subsequently, 550 (20.9%) deaths occurred within the time frame of 1 to 2 years, followed by 660 (25.1%) deaths within 2 to 5 years, and 357 (13.6%) deaths transpired beyond 5 years from the point of diagnosis. Among the included patients in the study, 665 (25.3%) died due to GC, whereas 1251 (47.6%) were attributed to non-gastric cancer causes. Additionally, 714 (27.1%) were accounted for by non-cancer-related factors. Notably, the occurrence of deaths stemming from non-cancer causes exceeded those attributed to gastric cancer by a margin of 2%.

Within the context of NCGC, the highest number of deaths (1102; 40.1%) occurred in the first year following the diagnosis of GC. This pattern was followed by 447 (16.2%) deaths transpiring within the interval of 1 to 2 years, 629 (22.9%) deaths transpiring within 2 to 5 years, and 573 (20.8%) deaths occurring more than 5 years after diagnosis. Among the analyzed patients, 1490 (54.2%) died due to GC, while 242 (8.8%) were attributed to non-gastric cancer causes. Furthermore, 1019 (37%) were attributable to non-cancer-related factors. Notably, the prevalence of deaths stemming from non-cancer causes was slightly lower than that attributed to gastric cancer by a margin of 17.2%.

### 3.2. Cancer-Related Causes of Death in Cardia Cancer

The prevailing non-gastric cancer cause was esophageal cancer [1094 patients; 87.5%, SMR = 259.48 (244.33–275.33)], followed by bronchogenic cancer [34; 2.7%, SMR = 0.94 (0.65–1.31)], miscellaneous malignancies [35; 2.8%, SMR = 3.43 (2.39–4.76)], and other digestive organs [10; 0.8% SMR = 34.14 (16.37–62.78)].

Among patients diagnosed with cardia cancer, noteworthy mortality risks were evident for esophageal cancer, miscellaneous malignancies, and other malignancies of the digestive organs. However, the mortality risk associated with bronchogenic cancer was found to be statistically insignificant. It is important to note that the mortality rate for bronchogenic cancer was lower than the expected rate in the general population.

### 3.3. Non-Cancer-Related Causes of Death in Cardia Cancer

Among patients diagnosed with GC, the foremost non-cancer cause of death was attributed to cardiovascular diseases [238; 33.3%, SMR = 1.49 (1.31–1.69)], followed by chronic obstructive pulmonary disease (COPD) and allied conditions [80; 11.2%, SMR = 2.19 (1.74–2.73)], accidents and adverse effects [33; 4.6%, SMR = 1.96 (1.35–2.76)], cerebrovascular diseases [32; 4.5%, SMR = 1.07 (0.73–1.51)], nephritis, nephrotic syndrome and nephrosis [27; 3.8%, SMR = 2.38 (1.57–3.46)], diabetes mellitus [26, 3.6%, SMR = 1.55 (1.01–2.27)], Alzheimer’s disease (ICD-9 and 10 only) [21, 2.9%, SMR = 0.98 (0.61–1.5)], septicemia [20; 2.8%, SMR = 2.52 (1.54–3.89), suicide and self-inflicted injury [16; 2.2%, SMR = 3.98 (2.1–5.98), other infectious and parasitic diseases including HIV [11; 1.5%, SMR = 2.46 (1.23–4.41), and chronic liver disease and cirrhosis [11; 1.5%, SMR = 1.89 (0.94–3.37). Detailed data are provided in Table 2.

In the initial year following diagnosis, the primary contributors to mortality were cardiovascular disease, trailed by COPD and allied conditions, septicemia, cerebrovascular diseases, suicide and self-inflicted injury, diabetes mellitus, and accidents and adverse effects. Among patients afflicted with cardia cancer, a significant mortality risk was discernible for various causes, including cardiovascular diseases, COPD and allied conditions, accidents and adverse effects, cerebrovascular diseases, nephritis, nephrotic syndrome and nephrosis, diabetes mellitus, septicemia, suicide and self-inflicted injury, as well as other infectious and parasitic diseases, including HIV. Conversely, the risk of death attributed to Alzheimer’s disease (ICD-9 and 10 only) was found to be statistically insignificant, with the mortality rate falling below the anticipated level within the general population. Conversely, though the mortality rate stemming from chronic liver disease and cirrhosis was recorded, the associated risk of death was found to be statistically insignificant. This trend may arise from factors such as the sample size or other considerations.

Regarding non-cancer causes, the most substantial surge in deaths (213; 29.8%) transpired within the initial year following the diagnosis of GC. Subsequently, 107 (15%) deaths transpired within the interval of 1 to 2 years, followed by 196 (27.5%) deaths within 2 to 5 years, and 198 (27.7%) deaths occurring more than 5 years after diagnosis. Remarkably, the occurrence of deaths decreased during the 1- to 2-year interval, only to exhibit an increase again after the 5-year mark.

### 3.4. Cancer Causes of Death in Non-Cardia Cancer

The most prevalent non-gastric cancer causes encompassed miscellaneous malignant tumors [44; 18.2%, SMR = 2.96 (2.15–3.98)]. Additionally, lung and bronchus cancers followed [38; 15.7%, SMR = 0.78 (0.55–1.07)], along with pancreatic cancers [33; 13.6%, SMR = 2.47 (1.7–3.47)], esophageal cancers [24; 9.9%, SMR = 5.53 (3.54–8.22)], and cancers of the colon and rectum [17; 7%, SMR = 0.94 (0.55–1.51)]. Moreover, other malignancies about the digestive organs contributed [14; 5.8%, SMR = 29.29 (16.02–49.15)]. Among patients affected by non-cardia cancer, the mortality risk attributed to miscellaneous malignant tumors, pancreatic cancer, esophageal cancer, and other malignancies of the digestive organs emerged as statistically significant. However, the risk of death linked to lung and bronchus cancer, colon cancer, and rectum cancer was observed to be statistically insignificant within this subgroup. Additionally, it is noteworthy that the mortality rates for lung and bronchus cancer, colon cancer, and rectum cancer were observed to be lower than that anticipated within the general population.

### 3.5. Non-Cancer-Related Causes of Death in Non-Cardia Cancer

The prevailing causes were predominantly cardiovascular diseases [375; 36.8%, SMR = 1.31 (1.18–1.45)], followed by cerebrovascular diseases [80; 7.9%, SMR = 1.25 (0.99–1.56)], COPD and allied conditions [68; 6.7%, SMR = 1.25 (0.97–1.58)], pneumonia and influenza [43; 4.2%, SMR = 1.59 (1.15–2.14)], diabetes mellitus [40, 3.9%, SMR = 1.32 (0.94–1.79)], accidents and adverse effects [38, 3.7%, SMR = 1.45 (1.03–1.99)], nephritis, nephrotic syndrome, and nephrosis [37; 3.6%, SMR = 1.69 (1.19–2.33)], and Alzheimer’s disease (ICD-9 and 10 only) [34, 3.3%, SMR = 0.69 (0.48–0.96)].

During the initial year following diagnosis, the principal causes were cardiovascular disease, followed by COPD and allied conditions, cerebrovascular diseases, and diabetes mellitus. Within the cohort of patients with non-cardia cancer, notable mortality risks were apparent for cardiovascular diseases, pneumonia and influenza, accidents, and adverse effects, as well as nephritis, nephrotic syndrome, and nephrosis. Conversely, the risk of death stemming from Alzheimer’s disease (ICD-9 and 10 only) was statistically significant; however, the associated mortality rate was observed to be below the anticipated level within the general population. Conversely, the risk of death related to cerebrovascular diseases, COPD and allied conditions, and diabetes mellitus was determined to be statistically insignificant.

Concerning non-cancer causes, the most substantial surge in deaths (262; 25.7%) occurred during the initial year following GC diagnosis. Subsequently, 106 (10.4%) deaths were recorded within the 1- to 2-year interval, with 250 (24.5%) transpiring within 2 to 5 years, followed by 401 (39.4%) deaths transpiring after more than 5 years post-diagnosis. Intriguingly, the occurrence of deaths demonstrated a decrease during the 1- to the 2-year interval, only to rise again after the 5-year mark. Comprehensive data can be found in Table 3.

While the mortality rate for Alzheimer’s disease (ICD-9 and 10 only) and chronic liver disease and cirrhosis was statistically insignificant in cardia cancer, it emerged as a significant cause in non-cardia cancer cases. Moreover, in cardia cancer, diabetes mellitus also surfaced as a significant cause.

Appendix A accompanying this article provides a comprehensive set of 42 tables presenting detailed findings. These tables focus on standardized mortality ratios (SMRs) for non-cancer causes in both cardia and non-cardia patients across various demographic and clinical subgroups. The tables cover different age groups (<39 years, 40–65 years, >65 years), gender (male, female), race (white, black, American Indian/Alaska Native, Asian/Pacific Islander, unknown race), time periods (2000–2004, 2005–2009, 2010–2014, 2015–2019), marital status (married, unmarried, unknown), and treatment modalities (surgical, radiotherapy, chemotherapy). Additionally, two tables examine SMRs specifically related to cancer causes for cardia and non-cardia patients. The information presented in these tables contributes to a comprehensive understanding of mortality patterns and associated factors in both cardia and non-cardia patients, offering valuable insights for further research and clinical decision-making.

### 3.6. Relative Survival and Conditional Survival Analysis

The 1-year and 5-year relative survival for CGC patients were 76.4%, 95%CI (75–77.7), and 48.9%, 95%CI (47–50.7), respectively, while for NCGC were 80.4%, 95%CI (79.2–81.6), and 63.9%, 95%CI (62.1–65.6). The 1-year and 5-year observed survival for CGC patients was 74.1%, 95%CI (72.7–75.4), and 42.1%, 95%CI (40.6–43.7), respectively, while for NCGC were 77.7%, 95%CI (76.6–78.8), and 53.7 95%CI (52.2–55.1). NCGC patients had better survival rates than CGC patients. The 3-year conditional survival rates for CGC patients were 68.7% (95% CI: 66.7–70.6) at 1 year and 88.8% (95% CI: 85.8–91.2) at 5 years. For NCGC patients, the rates were 82.2% (95% CI: 80.5–83.7) at 1 year and 93.5% (95% CI: 91.2–95.3) at 5 years. This indicates that the longer a person survives after cancer diagnosis, the greater their chances of surviving for another 3 years or more. For CGC patients, the likelihood of survival increased by at least 20 percentage points between 1 and 5 years’ post-diagnosis, while for NCGC patients, the increase was around 10%. This difference can serve as an indicator of CGC prognosis and the effectiveness of the treatments provided.

Additional results regarding survival rates categorized by different subgroups, including age (less than 50 years old and over 50 years old) and gender (male and female), can be found in Table 4 and Table 5.

### 3.7. Causes of Death in Cardia and Non-Cardia Cancer Patients in the Surgical Treatment Group

Among patients diagnosed with CGC and got surgical treatment, the foremost cause of death was attributed to non-GC [545; 45.9%, SMR = 5.43 (95% CI: 4.98–5.9)], followed by non-cancer causes [433; 36.6%, SMR 1.36 (95% CI: 1.23–1.49)] and gastric cancer [206; 17.4%, SMR = 104.95p (95% CI: 91.1–120.3)] (Table 6). The foremost non-cancer cause of death was attributed to cardiovascular diseases [117; 16.1%, SMR = 1 (95% CI: 0.83–1.2)], followed by chronic obstructive pulmonary disease (COPD) and allied conditions [43; 5.9%, SMR = 1.55 (95% CI: 1.12–2.09)], and cerebrovascular diseases [22; 3.1%, SMR = 1.01 (95% CI: 0.63–1.53)]. Detailed data for all causes is provided in Table 6.

When it comes to patients diagnosed with NCGC and got surgical treatment, the foremost cause of death was attributed to non-cancer causes [819; 50.4%, SMR = 1.17 (95% CI: 1.09–1.25)], followed by gastric cancer [636; 39.1%, SMR = 133.42 (95% CI: 123.25–144.21)] and non-GC causes [171; 4.27%, SMR = 1 (95% CI: 0.85–1.16)] (Table 7). Among patients with non-cancer causes, the top three leading causes of death were cardiovascular diseases [292; 36.5%, SMR = 1.14 (95% CI: 1.01–1.28)], followed by cerebrovascular diseases [64; 8.0%, SMR = 1.12 (95% CI: 0.86–1.43)] and COPD and allied conditions [49; 6.1%, SMR = 0.99 (95% CI: 0.74–1.31)]. Detailed data for all causes are provided in Table 7.

### 3.8. Causes of Death in Cardia and Non-Cardia Cancer Patients in the Radiotherapy Treatment Group

Among patients diagnosed with CGC and those who received radiotherapy treatment, the foremost cause of death was attributed to non-GC [684; 42.7%, SMR = 17.78 (95% CI: 16.48–19.17)], followed by non-cancer causes [238; 14.9%, SMR = 1.90 (95% CI: 1.67–2.16)] and GC [231; 14.4%, SMR = 302.34 (95% CI: 264.6–343.94)] (Table 8). Among patients with non-cancer causes, the top three leading causes of death were cardiovascular diseases [87; 10.7%, SMR = 1.85 (95% CI: 1.48–2.28)], followed by chronic obstructive pulmonary disease and allied conditions [29; 3.6%, SMR = 2.73 (95% CI: 1.83–3.93)] and pneumonia and influenza [11; 1.4%, SMR = 2.92 (95% CI: 1.46–5.23)]. Detailed data for all causes are provided in Table 8.

Among patients diagnosed with NCGC and those who received radiotherapy treatment, the foremost cause of death was attributed to GC [419; 34.6%, SMR = 532.43p (95% CI: 482.67–585.94)], followed by non-cancer causes [111; 9.2%, SMR = 1.1 (95% CI: 0.9–1.32)] and non-GC [37; 3.1%, SMR = 1.16 (95% CI: 0.82–1.6)] (Table 9). Among patients with non-cancer causes, the top three leading causes of death were cardiovascular diseases [29; 3.6%, SMR = 1.14 (95% CI: 0.77–1.64)], followed by cerebrovascular diseases [28; 3.4%, SMR = 1.52 (95% CI: 0.66–3)] and chronic obstructive pulmonary disease and allied conditions [8; 1.0%, SMR = 1.5 (95% CI: 0.65–2.95)]. Detailed data for all causes is provided in Table 9.

### 3.9. Causes of Death in Cardia and Non-Cardia Cancer Patients in the Chemotherapy Treatment Group

Among patients diagnosed with CGC and who received chemotherapy treatment, the foremost cause of death was attributed to Non-GC [673; 45.9%, SMR = 15.21p (95% CI: 14.09–16.41)], followed by gastric cancer [328; 22.4%, SMR = 373.18p (95% CI: 333.88–415.83)] and non-cancer causes [253; 17.2%, SMR = 1.85p (95% CI: 1.63–2.09)] (Table 10). Among patients with non-cancer causes, the top three leading causes of death were cardiovascular diseases [92; 36.5%, SMR = 1.81p (95% CI: 1.46–2.21)], followed by chronic obstructive pulmonary disease and allied conditions [32; 12.8%, SMR = 2.69p (95% CI: 1.84–3.8)] and accidents and adverse effects [13; 5.2%, SMR = 2.31p (95% CI: 1.23–3.95)]. Detailed data for all causes is provided in Table 10.

When it comes to the patients diagnosed with NCGC and received chemotherapy treatment, the foremost cause of death was attributed to GC [419; 50.4%, SMR = 532.43p (95% CI: 482.67–585.94)], followed by non-cancer causes [111; 13.5%, SMR = 1.1 (95% CI: 0.9–1.32)], followed by non-GC causes [37; 4.5%, SMR = 1.16 (95% CI: 0.82–1.6)] (Table 11). Among patients with non-cancer causes, the top three leading causes of death were cardiovascular diseases [34; 0.92 (95% CI: 0.64–1.29)], followed by chronic obstructive pulmonary disease and allied conditions [11; 1.37 (95% CI: 0.68–2.45)] and cerebrovascular diseases [8; 1.06 (95% CI: 0.46–2.08)]. Detailed data for all causes is provided in Table 11.

## 4. Discussion

GC is an important cancer to study, responsible for more than one million new cases and about 769,000 deaths in 2020 globally. Among all cancers, it is ranked fifth in incidence and fourth in mortality worldwide [19]. To our knowledge, this is the first comprehensive study to investigate the causes of death in early GC in terms of cardia and non-cardia causes of death.

We found that GC of both types occurs predominantly in patients who are white, male, and over 50 years old, findings consistent with previous studies [1,10,20]. Sung et al. (2021) reported a twofold increased incidence rate in males compared to females [19]. Many studies reported that throughout the past few decades, the incidence of cardia gastric cancer was increasing while that of non-cardia cancer was decreasing in Western countries, specifically the USA. Lv et al. (2021) reported that cardia cancer increased by 5% between 2004 and 2015 [1,4,11,12].

Other studies have reported a decline in the incidence of gastric cancer (GC) in the USA. Liu et al. (2022) found that the overall GC incidence decreased between 1992 and 2019, primarily due to a 1.48% annual decline in non-cardia GC, compared to a 0.52% annual decrease in cardia GC during the same period [6].

Another study by Wang et al. (2018) reported that the overall age-adjusted incidence rate of GC has decreased by 1.55% annually between 1999 and 2007 and remained unchanged between 2007 and 2013. In people aged ≥ 50 years this was due to a decrease in the incidence of non-cardia GC, as the incidence of cardia GC remained unchanged for those patients [21]. The discrepancy between studies should be investigated nationally and geographically as there is increasing evidence that there is a geographical impact on the rates of GC independent of race and ethnicity [10]. Of the included patients, 5381 died during the follow-up period; the highest number of deaths in both groups was during the first year after diagnosis.

Deaths from non-GC cancers were 47.6% in the cardia cancer group and 54.2% in the non-cardia cancer group. The most common non-GC cancer cause of death in the cardia cancer group was esophageal cancer. The most common non-GC cancer cause of death in the non-cardia cancer group was miscellaneous malignant diseases.

Esophageal cancer accounted for only 9.9% of non-GC deaths in the non-cardia cancer group. Several epidemiological studies have shown a correlation between gastric cardia adenocarcinoma and symptomatic gastro-esophageal reflux, although this link is less pronounced than that with distal esophageal adenocarcinoma [22]. Evidence suggests that adenocarcinoma of the gastric cardia originates from intestinal metaplasia of the cardia and shares similarities with distal esophageal metaplasia [23,24]. any researchers believe that cardia cancer belongs to the esophageal adenocarcinoma spectrum as it occurs within 3 cm below the gastroesophageal junction in the gastric mucosa and may arise from the short segment of the Barrett esophagus [25,26].

The most common non-cancer cause of death was cardiovascular diseases. This is consistent with previous studies that reported that cardiac death is the most important cause of death in cancer patients, even more important than the cancer itself after 10 years of diagnosis [27]. It is also well known that cancer treatments like chemotherapy and radiotherapy could induce cardiac injury as a side effect [28]. It is worth mentioning that our data did not include possible confounding variables in the studied population such as smoking, alcohol consumption, and lipid profile, which may influence any observed association between GC and cardiovascular diseases.

COPD disease and allied conditions was the second most common non-cancer cause of death in the cardia cancer group, while in the non-cardia group, it was the third most common non-cancer cause of death. Gastrointestinal tract malignancy and COPD share many risk factors like smoking, alcohol, and others. GC is one of the most frequent malignancies overrepresented in COPD (29). Also, COPD may be a relevant cause of complication development related to anti-cancer therapy [28,29]. A previous study reported that GC had a COPD prevalence of 13.4% [30].

Cerebrovascular diseases were the fourth most common cause of death in the CGC group and the second most common cause of death in the NCGC group. However the results were insignificant in the non-cardia cancer group. In our study, cerebrovascular disease as a cause of death was slightly higher in GC patients than in the general population. In other studies, it was lower in cancer groups compared to the general population [31,32]. Cerebrovascular disease represented about 16.8% of non-cancer causes of death in GC in a previous study, while in our study it represents about 10.3% [31]. The differences between studies could be attributed to geographical and demographic factors.

Other non-cancer causes include septicemia, pneumonia and influenza, nephritis, nephrotic syndrome, and nephrosis as well as diabetes, suicide and self-inflicted injuries, Alzheimer’s disease, infectious and parasitic diseases including HIV, and chronic liver disease and cirrhosis. The small number of those populations in our study prevented proper appraisal and interpretation with limited evidence.

Suicide and self-inflicting injuries represented 2.2% of deaths in the cardia cancer group, which is higher than the general population. Previous studies also reported increased SMR for suicide, especially in stomach cancer survivors and men [31,33,34].

The WHO defines early GC as the GC limited to the mucosa or sub-mucosa regardless of lymph node metastasis [12,35]. The 5-year survival rate of early GC after operation can reach about 90–95% [35]. In Japan, the 5-year survival rate of early GC is more than 90% while it is about 14–25% in advanced GC cases [36,37]. Most of the patients are diagnosed in the late or middle stage with high mortality and short survival rates and almost no hope for cure [38]. Early diagnosis and surgical intervention are the best treatment for patients with GC, which significantly prolongs their survival [39,40]. In light of the following facts, the early stage of GC is the decision-making stage in which diagnosis and curative treatment mostly matter. Therefore, we conducted our analysis on patients in stage I according to TNM staging based on AJCC sixth TNM.

The analysis of survival outcomes among patients with CGC and NCGC reveals insightful patterns in our study. Initially, the 1-year and 5-year relative survival rates portray a somewhat lower survival likelihood for CGC patients compared to NCGC patients. These results align within the results reported to the literature by Lv et al. [1].

The intriguing aspect arises when we consider 3 years conditional survival after 1 year and 5 years. For CGC patients, the conditional survival rates notably increase while NCGC patients demonstrate higher conditional survival rates, after 1 year and an impressive increase after 5 years. This highlights a key observation: the longer a person has already survived after their cancer diagnosis, the more optimistic the outlook for an additional 3 years or more, especially for NCGC patients.

Delving deeper into the dynamics of survival trends, the substantial 20-percentage-point increase in the likelihood of survival for CGC patients between 1 year and 5 years post-diagnosis is a noteworthy finding. In contrast, NCGC patients experienced a more modest increase, around 10%, during the same period. This discrepancy in the rate of increase points towards a potential indicator of CGC prognosis.

The traditional biomarkers as CEA and CA 19-9 play an important role in monitoring gastric cancer post resection, but their sensitivity in screening for the disease is usually insufficient [41,42,43]. By gene expression regulation, Micro-RNAs play a major role in gastric neoplasms. Their tissue specificity makes them reliable as biomarkers for tumor progression [44,45,46]. Previous evidence showed that miR-106 levels are elevated in the gastric juice of patients diagnosed with gastric cancer, which makes it a promising biomarker for diagnosis and follow up of gastric cancer [47]. Boicean et al. (2024) found that although gastric juice miR-106 is a good marker in detecting gastric adenocarcinoma, it is still inferior to the diagnostic capabilities of CEA and CA 19-9, which clear the need for more research in this area to improve early diagnosis [48].

Because the symptoms associated with gastric cancer can present with a wide range of pathologies, the diagnostic studies are a corner element in the diagnosis of gastric cancer with many emerging imaging methods. The endoscopic ultrasound was presented by many studies as a tool that can exclude malignancy and differentiate between the various pathologies with safety and efficacy higher to other modalities as upper endoscopy, CT, and exploratory laparotomy [49,50].

This study identified a lack of information in the literature about the differences in causes of death between the two groups, which requires further research to identify the causes of these differences and more personalized follow-up and treatment plans. In addition, our study is a population-based study with the data collected prospectively and independently of our study with a low risk of recall bias and a large sample size obtained from a high-quality (SEER) database, which includes 28% of USA cancer patients.

The lack of information about the patient-specific associated risk factors, the retrospective design, missing data, detailed treatment information, including specifics on endoscopic resections and resection margin status, and potential coding errors may increase the risk of bias in this study and prevent us from providing direct evidence about the role of specific exposures. The associated risk factors should be investigated to study their impact, including H-pylori, Gastro-esophageal reflux disease, alcoholism, and other esophageal and gastric diseases. In addition, we cannot be sure about the quality of every medical registrar who documented the cause of death in every patient. However, we believe our large population from a high-quality SEER database, which includes 28% of USA cancer patients, is generalizable.

We recommend further research regarding the differences between the two groups, the causes of these differences, and the development of tailored, more specific diagnosis, follow-up, and treatment plans according to the differences between the two groups, which will promote those patients’ survival and prognosis.

## 5. Conclusions

In conclusion, this study revealed that 55.4% of the cohort died during the follow-up period, with the highest mortality rate occurring within the first year post-diagnosis. Notably, NCGC patients had better survival rates than CGC patients, with 1-year and 5-year relative survival rates of 80.4% and 63.9%, respectively, compared to 76.4% and 48.9% for CGC patients. This study also highlights the importance of both cancer-related and non-cancer-related factors in the mortality of gastric cancer patients and sheds light on the substantial impact of non-cancer causes of death in GC patients, underscoring the necessity of considering comorbidities in their comprehensive management and follow-up.

## Figures and Tables

**Figure 1 cancers-16-04262-f001:**
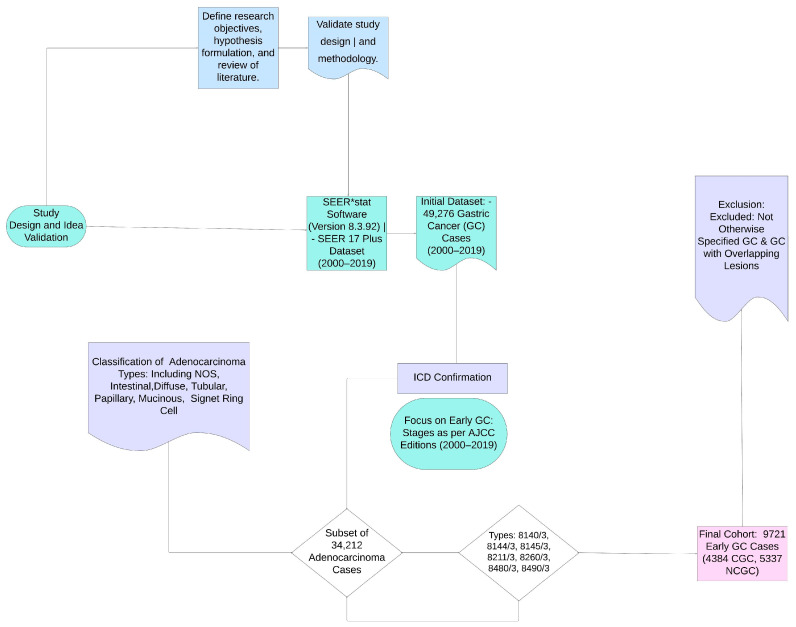
Flowchart of Study Design and Data Processing. This flowchart outlines the research process for this study using data from the SEER database. It includes steps for study design validation, data sourcing, initial dataset extraction, ICD code confirmation for adenocarcinoma cases, classification of adenocarcinoma types, focus on early-stage GC, application of exclusion criteria, and the final cohort selection.

**Figure 2 cancers-16-04262-f002:**
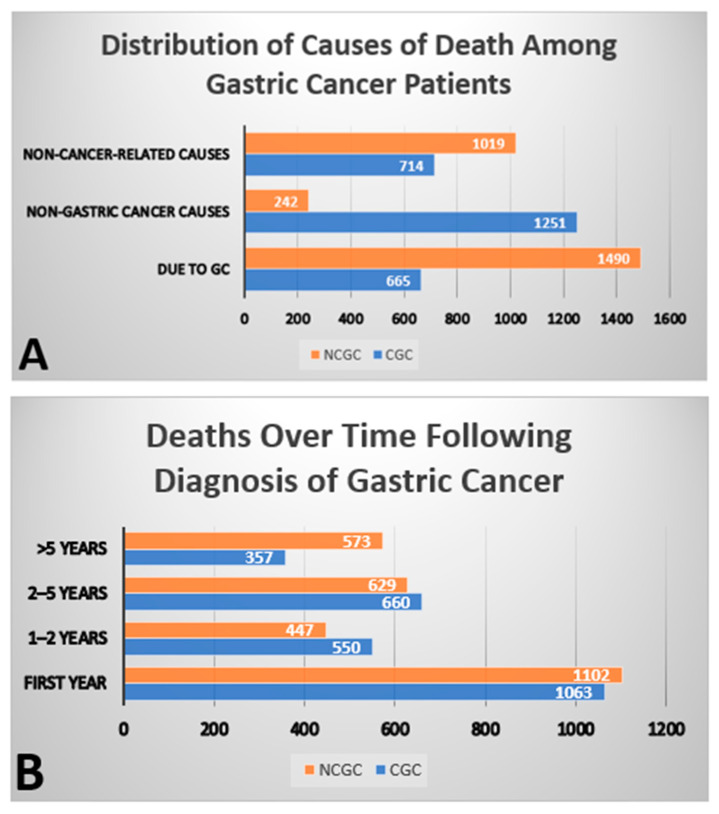
Distribution of Deaths in GC and NCGC Patients by Cause (**A**) and Time Interval from Diagnosis (**B**). The figure illustrates the percentage distribution of deaths in patients diagnosed with gastric cancer (GC) and non-cancerous gastric conditions (NCGC). Data are shown by the time interval from diagnosis: within 1 year, 1–2 years, 2–5 years, and beyond 5 years.

**Table 1 cancers-16-04262-t001:** Comparative Analysis of Demographic Characteristics Between Included Cardia and Non-Cardia Patients.

Variations	Cardia			Non-Cardia		
	Total			Total		
	Death	Patients	Mean Age at Event	Death	Patients	Mean Age at Event
Overall	2630 (60%)	4384	74.37	2751 (51.5%)	5337	78.75
**Age**						
<39	19 (35.8%)	53	38.76	23 (24.0%)	96	37.79
40–65	697 (45.8%)	1521	60.28	473 (17.2%)	1493	59.41
>65	1914 (68.1%)	2810	79.85	2255 (58.0%)	3748	83.22
**Sex**						
Male	2039 (59.6%)	3419	73.44	1505 (54.7%)	2889	77.56
Female	591 (61.3%)	965	77.57	1246 (34.7%)	2448	80.18
**Marital Status**						
Married	1502 (55.7%)	2696	73.86	1316 (46.5%)	2885	77.27
Unmarried	1016 (68.9%)	1475	75.04	1304 (55.6%)	2198	80.24
**Race**						
White	2324 (59.8%)	3886	74.31	1693 (47.3%)	3044	79.03
Black	144 (71.0%)	203	72.51	469 (57.6%)	814	75.07
Other	162 (54.9%)	295	76.85	589 (39.8%)	1479	80.86

**Table 2 cancers-16-04262-t002:** Standardized Mortality Ratios (SMRs) for Non-Cancer Causes in Cardia Patients: Comprehensive Analysis Across All Patient Groups.

Causes	<1 Years		1–2 Years		2–5 Years		>5 Years		Total		Total	
	Observed (n)	SMR (95%CI)	Observed (n)	SMR (95%CI)	Observed (n)	SMR (95%CI)	Observed (n)	SMR (95%CI)	Observed (n)	SMR (95%CI)	Patients	Mean Age at Event
Non-GC	503	18.24^p^ (16.68–19.9)	289	14.75^p^ (13.09–16.55)	336	8.52^p^ (7.63–9.48)	123	2.92^p^ (2.43–3.49)	1251	9.72^p^ (9.19–10.27)	4384	72.93
GC	347	609.40^p^ (546.96–677.01)	154	383.57^p^ (325.39–449.17)	128	163.53^p^ (136.43–194.44)	36	45.34^p^ (31.75–62.76)	665	261.02^p^ (241.56–281.64)	4384	74.36
All Causes of Death	1063	9.00^p^ (8.46–9.55)	550	6.70^p^ (6.15–7.28)	660	3.92^p^ (3.62–4.23)	357	1.85^p^ (1.67–2.06)	2630	4.68^p^ (4.51–4.87)	4384	74.37
Non-cancer causes	213	2.37^p^ (2.06–2.71)	107	1.72^p^ (1.41–2.08)	196	1.53^p^ (1.32–1.76)	198	1.32^p^ (1.14–1.52)	714	1.66^p^ (1.54–1.79)	4384	76.9
Septicemia	14	8.37^p^ (4.58–14.05)	1	0.85 (0.02–4.75)	3	1.25 (0.26–3.65)	2	0.74 (0.09–2.67)	20	2.52^p^ (1.54–3.89)	4384	72.56
Other Infectious and Parasitic Diseases including HIV	2	2.03 (0.25–7.35)	4	5.74^p^ (1.56–14.68)	2	1.45 (0.18–5.22)	3	2.14 (0.44–6.26)	11	2.46^p^ (1.23–4.41)	4384	71.58
Diabetes Mellitus	8	2.25 (0.97–4.44)	4	1.59 (0.43–4.08)	6	1.18 (0.43–2.56)	8	1.42 (0.61–2.8)	26	1.55^p^ (1.01–2.27)	4384	74.61
Alzheimer’s (ICD-9 and 10 only)	2	0.48 (0.06–1.74)	2	0.72 (0.09–2.6)	10	1.63 (0.78–2.99)	7	0.84 (0.34–1.72)	21	0.98 (0.61–1.5)	4384	84.9
Cardiovascular Diseases	79	2.30^p^ (1.82–2.87)	22	0.94 (0.59–1.43)	68	1.43^p^ (1.11–1.81)	69	1.27 (0.99–1.61)	238	1.49^p^ (1.31–1.69)	4384	79.53
Cerebrovascular Diseases	10	1.56 (0.75–2.87)	5	1.16 (0.38–2.7)	7	0.79 (0.32–1.63)	10	0.96 (0.46–1.77)	32	1.07 (0.73–1.51)	4384	80.23
Pneumonia and Influenza	5	1.77 (0.57–4.12)	4	2.11 (0.57–5.39)	6	1.56 (0.57–3.4)	4	0.92 (0.25–2.36)	19	1.47 (0.89–2.3)	4384	72.94
Chronic Obstructive Pulmonary Disease and Allied Cond	19	2.50^p^ (1.51–3.91)	17	3.18^p^ (1.85–5.09)	18	1.63 (0.97–2.58)	26	2.08^p^ (1.36–3.05)	80	2.19^p^ (1.74–2.73)	4384	77.46
Chronic Liver Disease and Cirrhosis	2	1.64 (0.2–5.94)	4	4.41^p^ (1.2–11.3)	3	1.63 (0.34–4.77)	2	1.07 (0.13–3.86)	11	1.89 (0.94–3.37)	4384	72.89
Nephritis, Nephrotic Syndrome and Nephrosis	4	1.66 (0.45–4.25)	6	3.60^p^ (1.32–7.83)	10	2.93^p^ (1.4–5.38)	7	1.82 (0.73–3.75)	27	2.38^p^ (1.57–3.46)	4384	75.85
Accidents and Adverse Effects	8	2.33^p^ (1.01–4.59)	5	2.05 (0.67–4.79)	9	1.77 (0.81–3.36)	11	1.88 (0.94–3.36)	33	1.96^p^ (1.35–2.76)	4384	70.57
Suicide and Self-Inflicted Injury	9	9.83^p^ (4.5–18.66)	2	2.97 (0.36–10.73)	1	0.73 (0.02–4.09)	4	2.87 (0.78–7.34)	16	3.68^p^ (2.1–5.98)	4384	66.09
Other Cause of Death	51	2.48^p^ (1.85–3.26)	31	2.17^p^ (1.47–3.08)	53	1.74^p^ (1.31–2.28)	45	1.21 (0.88–1.62)	180	1.76^p^ (1.51–2.03)	4384	75.74

^P^: means its significant (<0.05).

**Table 3 cancers-16-04262-t003:** Standardized Mortality Ratios (SMRs) for Non-Cancer Causes in Non-Cardia Patients: Comprehensive Analysis Across All Patient Groups.

Causes	<1 Years		1–2 Years		2–5 Years		>5 Years		Total		Total	
	Observed (n)	SMR (95%CI)	Observed (n)	SMR (95%CI)	Observed (n)	SMR (95%CI)	Observed (n)	SMR (95%CI)	Observed (n)	SMR (95%CI)	Patients	Mean Age at Event
Non-GC	60	1.82^p^ (1.39–2.34)	40	1.56^p^ (1.11–2.12)	58	1 (0.76–1.29)	84	1.19 (0.95–1.47)	242	1.29^p^ (1.13–1.46)	5337	78.14
GC	780	839.73^p^ (781.82–900.78)	301	412.62^p^ (367.32–461.97)	321	197.43^p^ (176.42–220.25)	88	45.87^p^ (36.79–56.51)	1490	286.38^p^ (272.02–301.3)	5337	76.64
All Causes of Death	1102	6.74^p^ (6.35–7.15)	447	3.51^p^ (3.2–3.86)	629	2.14^p^ (1.98–2.31)	573	1.47^p^ (1.35–1.6)	2751	2.82^p^ (2.72–2.93)	5337	78.75
Non-cancer causes	262	2.02^p^ (1.79–2.28)	106	1.05 (0.86–1.27)	250	1.07 (0.94–1.21)	401	1.27^p^ (1.15–1.4)	1019	1.30^p^ (1.23–1.39)	5337	81.97
Septicemia	11	4.51^p^ (2.25–8.07)	1	0.53 (0.01–2.96)	5	1.16 (0.38–2.71)	11	2 (1–3.57)	28	1.98^p^ (1.32–2.86)	5337	76.92
Other Infectious and Parasitic Diseases including HIV	6	4.39^p^ (1.61–9.56)	2	1.86 (0.23–6.73)	9	3.73^p^ (1.7–7.07)	3	1.05 (0.22–3.07)	20	2.60^p^ (1.59–4.01)	5337	75.2
Diabetes Mellitus	12	2.30^p^ (1.19–4.02)	6	1.48 (0.54–3.22)	9	0.97 (0.45–1.85)	13	1.1 (0.58–1.88)	40	1.32 (0.94–1.79)	5337	77.76
Alzheimer’s (ICD-9 and 10 only)	5	0.69 (0.22–1.61)	2	0.35 (0.04–1.26)	4	0.28^p^ (0.08–0.72)	23	1.04 (0.66–1.56)	34	0.69^p^ (0.48–0.96)	5337	90.38
Cardiovascular Diseases	101	2.06^p^ (1.68–2.51)	35	0.93 (0.65–1.29)	80	0.93 (0.74–1.16)	159	1.40^p^ (1.19–1.64)	375	1.31^p^ (1.18–1.45)	5337	83.57
Cerebrovascular Diseases	19	1.77^p^ (1.06–2.76)	6	0.72 (0.27–1.57)	24	1.26 (0.81–1.87)	31	1.21 (0.82–1.71)	80	1.25 (0.99–1.56)	5337	83.11
Pneumonia and Influenza	10	2.17^p^ (1.04–3.98)	5	1.4 (0.46–3.27)	15	1.84^p^ (1.03–3.04)	13	1.21 (0.65–2.08)	43	1.59^p^ (1.15–2.14)	5337	83.33
Chronic Obstructive Pulmonary Disease and Allied Cond	20	2.17^p^ (1.33–3.35)	8	1.11 (0.48–2.19)	22	1.33 (0.83–2.01)	18	0.84 (0.5–1.33)	68	1.25 (0.97–1.58)	5337	79.12
Chronic Liver Disease and Cirrhosis	6	5.45^p^ (2–11.85)	2	2.29 (0.28–8.28)	11	5.63^p^ (2.81–10.08)	5	2.15 (0.7–5.01)	24	3.83^p^ (2.46–5.71)	5337	68.24
Nephritis, Nephrotic Syndrome and Nephrosis	9	2.41^p^ (1.1–4.58)	5	1.73 (0.56–4.04)	9	1.35 (0.62–2.57)	14	1.62 (0.89–2.72)	37	1.69^p^ (1.19–2.33)	5337	82.72
Accidents and Adverse Effects	5	1.18 (0.38–2.76)	4	1.2 (0.33–3.07)	10	1.28 (0.61–2.35)	19	1.77^p^ (1.06–2.76)	38	1.45^p^ (1.03–1.99)	5337	78.09
Suicide and Self-Inflicted Injury	2	2.92 (0.35–10.56)	0	0 (0–6.82)	1	0.83 (0.02–4.64)	0	0 (0–2.55)	3	0.77 (0.16–2.26)	5337	74.25
Other Cause of Death	56	1.87^p^ (1.41–2.43)	30	1.27 (0.85–1.81)	51	0.9 (0.67–1.18)	92	1.15 (0.93–1.41)	229	1.20^p^ (1.05–1.37)	5337	82.27

^P^: means its significant (<0.05).

**Table 4 cancers-16-04262-t004:** The Observed Survival, Relative Survival, and Conditional Survival for Cardia Cancer.

Cumulative Summary/Age < 50 Years.					
Survival	N	Observed Survival	95%CI (Lower Limit–Upper Limit)	Relative Survival	95%CI (Lower Limit–Upper Limit)
1 Year	249	83.00%	(77.7–87.2)	83.30%	(77.9–87.5)
5 Years	249	61.70%	(54.9–67.7)	62.70%	(55.8–68.8)
3 Yrs Conditional Survival At 1 Year	193	77.30%	(70.5–82.8)	78.10%	(71.1–83.6)
3 Yrs Conditional Survival At 5 Years	118	93.70%	(87.3–97)	94.50%	(87.7–97.6)
**Cumulative Summary/Age > 50** **Years**					
1 Year	4153	73.50%	(72.1–74.9)	76.00%	(74.5–77.3)
5 Years	4153	40.90%	(39.3–42.6)	48.00%	(46.1–49.8)
3 Yrs Conditional Survival At 1 Year	2866	62.00%	(60.1–63.8)	68.00%	(65.9–70)
3 Yrs Conditional Survival At 5 Years	1179	79.00%	(76.3–81.5)	88.00%	(84.8–90.6)
**Cumulative Summary/Male**					
1 Year	3434	74.80%	(73.2–76.2)	77.10%	(75.5–78.6)
5 Years	3434	42.10%	(40.4–43.9)	49.00%	(46.9–51.1)
3 Yrs Conditional Survival At 1 Year	2417	62.70%	(60.6–64.6)	68.50%	(66.2–70.6)
3 Yrs Conditional Survival At 5 Years	1010	81.40%	(78.5–83.9)	89.90%	(86.5–92.5)
**Cumulative Summary/Female**					
1 Year	968	71.60%	(68.6–74.4)	73.80%	(70.7–76.7)
5 Years	968	42.10%	(38.7–45.4)	48.10%	(44.3–51.9)
3 Yrs Conditional Survival At 1 Year	642	64.30%	(60.3–68)	69.50%	(65.1–73.4)
3 Yrs Conditional Survival At 5 Years	287	77.60%	(71.7–82.4)	85.10%	(78.1–89.9)
**Cumulative Summary/All Patients**					
1 Year	4402	74.10%	(72.7–75.4)	76.40%	(75–77.7)
5 Years	4402	42.10%	(40.6–43.7)	48.90%	(47–50.7)
3 Yrs Conditional Survival At 1 Year	3059	63.00%	(61.2–64.8)	68.70%	(66.7–70.6)
3 Yrs Conditional Survival At 5 Years	1297	80.50%	(78–82.8)	88.80%	(85.8–91.2)

**Table 5 cancers-16-04262-t005:** The Observed Survival, Relative Survival, and Conditional Survival for Non-Cardia Cancer.

**Cumulative Summary/Age < 50 Years**					
Survival	N	Observed Survival	95%CI (Lower Limit–Upper Limit)	Relative Survival	95%CI (Lower Limit–Upper Limit)
1 Year	405	91.40%	(88.1–93.8)	91.50%	(88.2–94)
5 Years	405	75.50%	(70.6–79.8)	76.50%	(71.4–80.7)
3 Yrs Conditional Survival At 1 Year	337	86.10%	(81.8–89.5)	86.70%	(82.3–90.1)
3 Yrs Conditional Survival At 5 Years	204	93.80%	(89.1–96.5)	94.60%	(89.6–97.2)
**Cumulative Summary/Age > 50 Years**					
1 Year	4957	76.60%	(75.4–77.8)	79.50%	(78.3–80.7)
5 Years	4957	51.90%	(50.4–53.4)	62.80%	(61–64.6)
3 Yrs Conditional Survival At 1 Year	3533	73.00%	(71.4–74.5)	81.70%	(79.9–83.3)
3 Yrs Conditional Survival At 5 Years	1841	81.70%	(79.7–83.5)	93.40%	(90.8–95.2)
**Cumulative Summary/Male**					
1 Year	2909	77.70%	(76.1–79.2)	80.50%	(78.9–82.1)
5 Years	2909	53.10%	(51.1–55)	63.60%	(61.2–65.9)
3 Yrs Conditional Survival At 1 Year	2116	73.50%	(71.4–75.4)	81.70%	(79.4–83.8)
3 Yrs Conditional Survival At 5 Years	1116	83.30%	(80.8–85.6)	94.40%	(91–96.6)
**Cumulative Summary/Female**					
1 Year	2453	77.80%	(76–79.4)	80.30%	(78.5–82)
5 Years	2453	54.30%	(52.2–56.4)	64.20%	(61.6–66.7)
3 Yrs Conditional Survival At 1 Year	1754	74.90%	(72.7–76.9)	82.70%	(80.2–84.9)
3 Yrs Conditional Survival At 5 Years	929	82.40%	(79.5–84.9)	92.10%	(88.5–94.6)
**Cumulative Summary/All Patients**					
1 Year	5362	77.70%	(76.6–78.8)	80.40%	(79.2–81.6)
5 Years	5362	53.70%	(52.2–55.1)	63.90%	(62.1–65.6)
3 Yrs Conditional Survival At 1 Year	3870	74.10%	(72.6–75.5)	82.20%	(80.5–83.7)
3 Yrs Conditional Survival At 5 Years	2045	82.90%	(81.1–84.6)	93.50%	(91.2–95.3)

**Table 6 cancers-16-04262-t006:** Standardized Mortality Ratios (SMRs) for Non-Cancer Causes for Cardia in Surgical Treatment Patients.

Causes	<1 Year		1–2 Years		2–5 Years		>5 Years		Total		Total	
	Observed (n)	SMR (95%CI)	Observed (n)	SMR (95%CI)	Observed (n)	SMR (95%CI)	Observed (n)	SMR (95%CI)	Observed (n)	SMR (95%CI)	Patients	Mean Age at Event
Non-GC	107	6.60^p^ (5.41–7.97)	125	9.03^p^ (7.51–10.75)	208	6.48^p^ (5.63–7.42)	105	2.75^p^ (2.25–3.32)	545	5.43^p^ (4.98–5.9)	2707	70.14
GC	55	166.96^p^ (125.78–217.32)	54	192.93^p^ (144.93–251.73)	65	102.71^p^ (79.27–130.92)	32	44.40^p^ (30.37–62.67)	206	104.95^p^ (91.1–120.3)	2707	71.19
All Causes of Death	240	3.86^p^ (3.39–4.38)	237	4.40^p^ (3.86–4.99)	409	3.11^p^ (2.82–3.43)	298	1.72^p^ (1.53–1.93)	1184	2.81^p^ (2.66–2.98)	2707	72.68
Non-cancer causes	78	1.71^p^ (1.35–2.13)	58	1.46^p^ (1.11–1.89)	136	1.38^p^ (1.16–1.63)	161	1.20^p^ (1.02–1.4)	433	1.36^p^ (1.23–1.49)	2707	76.59
Septicemia	9	10.18^p^ (4.65–19.32)	1	1.29 (0.03–7.21)	2	1.07 (0.13–3.85)	2	0.82 (0.1–2.97)	14	2.35^p^ (1.28–3.94)	2707	73.67
Other Infectious and Parasitic Diseases including HIV	1	1.74 (0.04–9.68)	1	2.03 (0.05–11.33)	2	1.78 (0.22–6.44)	2	1.58 (0.19–5.69)	6	1.73 (0.64–3.78)	2707	73.75
Diabetes Mellitus	2	1 (0.12–3.61)	3	1.74 (0.36–5.09)	6	1.47 (0.54–3.2)	7	1.37 (0.55–2.83)	18	1.4 (0.83–2.21)	2707	73.43
Alzheimer’s (ICD-9 and 10 only)	0	0 (0–2.19)	1	0.65 (0.02–3.62)	6	1.36 (0.5–2.96)	5	0.67 (0.22–1.57)	12	0.8 (0.41–1.39)	2707	85.15
Cardiovascular Diseases	20	1.16 (0.71–1.79)	11	0.74 (0.37–1.32)	35	0.96 (0.67–1.34)	51	1.05 (0.78–1.38)	117	1 (0.83–1.2)	2707	78.9
Cerebrovascular Diseases	2	0.65 (0.08–2.33)	3	1.12 (0.23–3.26)	7	1.04 (0.42–2.15)	10	1.07 (0.51–1.97)	22	1.01 (0.63–1.53)	2707	81.33
Pneumonia and Influenza	3	2.29 (0.47–6.7)	2	1.74 (0.21–6.29)	6	2.1 (0.77–4.56)	3	0.77 (0.16–2.26)	14	1.52 (0.83–2.56)	2707	74.55
Chronic Obstructive Pulmonary Disease and Allied Cond	6	1.46 (0.53–3.17)	8	2.23 (0.96–4.38)	11	1.26 (0.63–2.26)	18	1.59 (0.94–2.51)	43	1.55^p^ (1.12–2.09)	2707	77.95
Chronic Liver Disease and Cirrhosis	1	1.19 (0.03–6.61)	0	0 (0–5.18)	1	0.63 (0.02–3.51)	2	1.16 (0.14–4.21)	4	0.82 (0.22–2.11)	2707	69.98
Nephritis, Nephrotic Syndrome and Nephrosis	2	1.68 (0.2–6.07)	4	3.82^p^ (1.04–9.77)	7	2.69^p^ (1.08–5.55)	6	1.74 (0.64–3.79)	19	2.29^p^ (1.38–3.58)	2707	77.9
Accidents and Adverse Effects	2	1.03 (0.12–3.71)	2	1.19 (0.14–4.3)	7	1.73 (0.69–3.56)	10	1.89 (0.91–3.48)	21	1.62^p^ (1–2.48)	2707	69.06
Suicide and Self-Inflicted Injury	4	6.46^p^ (1.76–16.53)	1	1.92 (0.05–10.7)	1	0.86 (0.02–4.79)	4	3.14 (0.86–8.04)	10	2.80^p^ (1.34–5.14)	2707	64.75
Other Cause of Death	26	2.58^p^ (1.68–3.78)	21	2.34^p^ (1.45–3.58)	45	1.95^p^ (1.42–2.61)	41	1.23 (0.89–1.67)	133	1.76^p^ (1.48–2.09)	2707	75.72

^P^: means its significant (<0.05).

**Table 7 cancers-16-04262-t007:** Standardized Mortality Ratios (SMRs) for Non-Cancer Causes for Non-Cardia in Surgical Treatment Patients.

Causes	<1 Year		1–2 Years		2–5 Years		>5 Years		Total		Total	
	Observed (n)	SMR (95%CI)	Observed (n)	SMR (95%CI)	Observed (n)	SMR (95%CI)	Observed (n)	SMR (95%CI	Observed (n)	SMR (95%CI	Patients	Mean Age at Event
Non-GC	11	0.43^p^ (0.22–0.77)	24	1.06 (0.68–1.58)	53	0.98 (0.73–1.28)	83	1.2 (0.96–1.49)	171	1 (0.85–1.16)	4004	78.48
GC	210	290.77^p^ (252.77–332.87)	123	192.24^p^ (159.77–229.37)	225	147.10^p^ (128.51–167.63)	78	41.59^p^ (32.88–51.91)	636	133.42^p^ (123.25–144.21)	4004	74.27
All Causes of Death	363	3.06^p^ (2.76–3.4)	227	2.11^p^ (1.84–2.4)	493	1.81^p^ (1.66–1.98)	543	1.43^p^ (1.31–1.56)	1626	1.85^p^ (1.76–1.95)	4004	78.66
Non-cancer causes	142	1.54^p^ (1.3–1.81)	80	0.95 (0.75–1.18)	215	1 (0.87–1.14)	382	1.24^p^ (1.12–1.37)	819	1.17^p^ (1.09–1.25)	4004	82.11
Septicemia	11	6.18^p^ (3.09–11.06)	1	0.62 (0.02–3.48)	5	1.25 (0.41–2.93)	11	2.05^p^ (1.02–3.67)	28	2.20^p^ (1.46–3.18)	4004	76.92
Other Infectious and Parasitic Diseases including HIV	5	4.78^p^ (1.55–11.16)	2	2.15 (0.26–7.77)	9	3.99^p^ (1.82–7.58)	3	1.08 (0.22–3.15)	19	2.71^p^ (1.63–4.23)	4004	76.01
Diabetes Mellitus	8	2.04 (0.88–4.01)	4	1.14 (0.31–2.92)	6	0.7 (0.26–1.51)	13	1.13 (0.6–1.93)	31	1.12 (0.76–1.59)	4004	79.24
Alzheimer’s (ICD-9 and 10 only)	1	0.21 (0.01–1.17)	0	0.00^p^ (0–0.8)	3	0.23^p^ (0.05–0.68)	22	1.02 (0.64–1.55)	26	0.59^p^ (0.39–0.87)	4004	90.65
Cardiovascular Diseases	49	1.41^p^ (1.04–1.86)	25	0.79 (0.51–1.17)	66	0.84 (0.65–1.06)	152	1.38^p^ (1.17–1.61)	292	1.14^p^ (1.01–1.28)	4004	83.49
Cerebrovascular Diseases	12	1.57 (0.81–2.74)	4	0.58 (0.16–1.48)	20	1.14 (0.7–1.76)	28	1.12 (0.74–1.62)	64	1.12 (0.86–1.43)	4004	82.67
Pneumonia and Influenza	7	2.15 (0.87–4.44)	4	1.35 (0.37–3.45)	12	1.6 (0.83–2.79)	13	1.24 (0.66–2.13)	36	1.49^p^ (1.04–2.06)	4004	83.16
Chronic Obstructive Pulmonary Disease and Allied Cond	8	1.17 (0.51–2.31)	7	1.14 (0.46–2.34)	19	1.23 (0.74–1.92)	15	0.72 (0.4–1.18)	49	0.99 (0.74–1.31)	4004	78.61
Chronic Liver Disease and Cirrhosis	1	1.1 (0.03–6.14)	1	1.27 (0.03–7.09)	9	4.87^p^ (2.23–9.24)	5	2.2 (0.71–5.12)	16	2.75^p^ (1.57–4.46)	4004	69.45
Nephritis, Nephrotic Syndrome and Nephrosis	4	1.49 (0.41–3.82)	5	2.05 (0.67–4.78)	9	1.46 (0.67–2.78)	13	1.55 (0.82–2.65)	31	1.58^p^ (1.07–2.24)	4004	82.67
Accidents and Adverse Effects	2	0.64 (0.08–2.31)	4	1.41 (0.38–3.6)	9	1.24 (0.57–2.35)	19	1.81^p^ (1.09–2.83)	34	1.43 (0.99–2)	4004	77.8
Suicide and Self-Inflicted Injury	2	3.59 (0.43–12.95)	0	0 (0–7.58)	1	0.88 (0.02–4.9)	0	0 (0–2.6)	3	0.83 (0.17–2.44)	4004	74.25

^P^: means its significant (<0.05).

**Table 8 cancers-16-04262-t008:** Standardized Mortality Ratios (SMRs) for Non-Cancer Causes for Cardia in Radiotherapy Patients.

Causes	<1 Year		1–2 Years		2–5 Years		>5 Years		Total		Total	
	Observed (n)	SMR (95%CI)	Observed (n)	SMR (95%CI)	Observed (n)	SMR (95%CI)	Observed (n)	SMR (95%CI)	Observed (n)	SMR (95%CI)	Patients	Mean Age at Event
Non-GC	244	22.74^p^ (19.98–25.78)	189	26.33^p^ (22.71–30.37)	210	18.07^p^ (15.7–20.68)	41	4.59^p^ (3.29–6.23)	684	17.78^p^ (16.48–19.17)	1603	71.8
GC	94	432.03^p^ (349.13–528.7)	66	454.63^p^ (351.61–578.4)	60	261.24^p^ (199.36–336.27)	11	64.09^p^ (31.99–114.68)	231	302.34^p^ (264.6–343.94)	1603	73.03
All Causes of Death	397	8.62^p^ (7.79–9.51)	303	9.92^p^ (8.84–11.11)	342	6.90^p^ (6.19–7.68)	111	2.90^p^ (2.38–3.49)	1153	7.01^p^ (6.61–7.43)	1603	72.76
Non-cancer causes	59	1.68^p^ (1.28–2.17)	48	2.07^p^ (1.52–2.74)	72	1.91^p^ (1.49–2.41)	59	2.02^p^ (1.54–2.6)	238	1.90^p^ (1.67–2.16)	1603	75.24
Septicemia	3	4.63 (0.96–13.54)	0	0 (0–8.53)	0	0 (0–5.21)	0	0 (0–6.75)	3	1.28 (0.26–3.76)	1603	69.08
Other Infectious and Parasitic Diseases including HIV	1	2.61 (0.07–14.53)	4	15.54^p^ (4.23–39.78)	1	2.42 (0.06–13.47)	1	3.38 (0.09–18.84)	7	5.18^p^ (2.08–10.68)	1603	68.19
Diabetes Mellitus	1	0.73 (0.02–4.07)	0	0 (0–4.02)	1	0.66 (0.02–3.69)	2	1.68 (0.2–6.06)	4	0.8 (0.22–2.05)	1603	78.62
Alzheimer’s (ICD-9 and 10 only)	0	0 (0–2.3)	1	0.95 (0.02–5.31)	2	1.16 (0.14–4.19)	1	0.71 (0.02–3.97)	4	0.69 (0.19–1.77)	1603	86.96
Cardiovascular Diseases	22	1.64^p^ (1.03–2.48)	10	1.14 (0.55–2.09)	31	2.21^p^ (1.5–3.13)	24	2.24^p^ (1.43–3.33)	87	1.85^p^ (1.48–2.28)	1603	78.13
Cerebrovascular Diseases	4	1.63 (0.44–4.17)	0	0 (0–2.31)	1	0.4 (0.01–2.21)	2	1.03 (0.12–3.72)	7	0.82 (0.33–1.69)	1603	78.36
Pneumonia and Influenza	4	3.62 (0.99–9.26)	4	5.64^p^ (1.54–14.43)	1	0.89 (0.02–4.97)	2	2.42 (0.29–8.75)	11	2.92^p^ (1.46–5.23)	1603	68.1
Chronic Obstructive Pulmonary Disease and Allied Cond	4	1.35 (0.37–3.45)	7	3.55^p^ (1.43–7.31)	8	2.51^p^ (1.08–4.94)	10	4.04^p^ (1.94–7.43)	29	2.73^p^ (1.83–3.93)	1603	76.9
Chronic Liver Disease and Cirrhosis	1	2.11 (0.05–11.76)	3	9.05^p^ (1.87–26.44)	2	3.52 (0.43–12.72)	0	0 (0–8.29)	6	3.30^p^ (1.21–7.18)	1603	76.35
Nephritis, Nephrotic Syndrome and Nephrosis	1	1.06 (0.03–5.9)	4	6.41^p^ (1.75–16.41)	3	2.97 (0.61–8.68)	4	5.21^p^ (1.42–13.33)	12	3.59^p^ (1.85–6.26)	1603	73.77
Accidents and Adverse Effects	3	2.22 (0.46–6.5)	3	3.3 (0.68–9.64)	4	2.61 (0.71–6.69)	3	2.49 (0.51–7.29)	13	2.60^p^ (1.39–4.45)	1603	71.68
Suicide and Self-Inflicted Injury	2	5.45 (0.66–19.69)	1	3.97 (0.1–22.1)	0	0 (0–8.66)	1	3.05 (0.08–17.01)	4	2.91 (0.79–7.46)	1603	62.19
Other Cause of Death	13	1.62 (0.86–2.77)	11	2.05^p^ (1.02–3.67)	18	2.02^p^ (1.2–3.19)	9	1.27 (0.58–2.42)	51	1.74^p^ (1.29–2.28)	1603	72.75

^P^: means its significant (<0.05).

**Table 9 cancers-16-04262-t009:** Standardized Mortality Ratios (SMRs) for Non-Cancer Causes for Non-Cardia in Radiotherapy Patients.

Causes	<1 Year		1–2 Years		2–5 Years		>5 Years		Total		Total	
	Observed (n)	SMR (95%CI)	Observed (n)	SMR (95%CI)	Observed (n)	SMR (95%CI)	Observed (n)	SMR (95%CI)	Observed (n)	SMR (95%CI)	Patients	Mean Age at Event
Non-GC	6	1.62 (0.59–3.52)	5	1.85 (0.6–4.31)	4	0.72 (0.2–1.85)	14	1.61 (0.88–2.7)	29	1.4 (0.94–2.01)	631	71.52
GC	74	770.43^p^ (604.95–967.21)	76	1099.03^p^ (865.91–1375.61)	76	538.15^p^ (424–673.57)	18	84.05^p^ (49.81–132.84)	244	468.71^p^ (411.74–531.35)	631	72.92
All Causes of Death	94	5.45^p^ (4.41–6.67)	97	8.13^p^ (6.6–9.92)	106	4.73^p^ (3.87–5.72)	63	1.62^p^ (1.25–2.07)	360	3.98^p^ (3.58–4.41)	631	73.79
Non-cancer causes	14	1.04 (0.57–1.75)	16	1.75 (1–2.84)	26	1.55^p^ (1.02–2.28)	31	1.04 (0.7–1.47)	87	1.26^p^ (1.01–1.55)	631	76.99
Septicemia	1	3.9 (0.1–21.71)	0	0 (0–20.71)	1	2.92 (0.07–16.28)	1	1.75 (0.04–9.75)	3	2.22 (0.46–6.5)	631	68.17
Other Infectious and Parasitic Diseases including HIV	0	0 (0–22.85)	0	0 (0–31.2)	2	8.48^p^ (1.03–30.65)	1	2.99 (0.08–16.68)	3	3.53 (0.73–10.32)	631	62.39
Diabetes Mellitus	0	0 (0–6.68)	2	5.14 (0.62–18.58)	1	1.29 (0.03–7.16)	1	0.77 (0.02–4.3)	4	1.33 (0.36–3.4)	631	73.31
Alzheimer’s (ICD-9 and 10 only)	0	0 (0–5.17)	0	0 (0–8.08)	0	0 (0–5.07)	0	0 (0–2.33)	0	0 (0–1.06)	631	
Cardiovascular Diseases	6	1.19 (0.44–2.59)	4	1.17 (0.32–3)	9	1.47 (0.67–2.79)	10	0.93 (0.44–1.7)	29	1.14 (0.77–1.64)	631	80.06
Cerebrovascular Diseases	1	0.95 (0.02–5.3)	1	1.42 (0.04–7.9)	2	1.59 (0.19–5.76)	4	1.78 (0.49–4.56)	8	1.52 (0.66–3)	631	79.05
Pneumonia and Influenza	0	0 (0–8.4)	0	0 (0–12.76)	0	0 (0–7.42)	1	1.1 (0.03–6.11)	1	0.47 (0.01–2.61)	631	79.42
Chronic Obstructive Pulmonary Disease and Allied Cond	1	1.02 (0.03–5.69)	2	2.9 (0.35–10.47)	3	2.23 (0.46–6.51)	2	0.86 (0.1–3.09)	8	1.5 (0.65–2.95)	631	76
Chronic Liver Disease and Cirrhosis	0	0 (0–24.9)	0	0 (0–31.19)	3	11.40^p^ (2.35–33.32)	0	0 (0–9.94)	3	3.33 (0.69–9.73)	631	71.03
Nephritis, Nephrotic Syndrome and Nephrosis	0	0 (0–9.75)	0	0 (0–14.29)	0	0 (0–7.65)	1	1.18 (0.03–6.58)	1	0.51 (0.01–2.83)	631	60.34
Accidents and Adverse Effects	1	2.06 (0.05–11.46)	1	2.86 (0.07–15.93)	3	4.36 (0.9–12.74)	2	1.7 (0.21–6.13)	7	2.59^p^ (1.04–5.34)	631	77.69
Suicide and Self-Inflicted Injury	0	0 (0–38.64)	0	0 (0–49.39)	0	0 (0–22.48)	0	0 (0–15.49)	0	0 (0–6.44)	631	
Other Cause of Death	4	1.28 (0.35–3.27)	6	2.84^p^ (1.04–6.18)	2	0.52 (0.06–1.9)	8	1.11 (0.48–2.18)	20	1.23 (0.75–1.9)	631	77.7

^P^: means its significant (<0.05).

**Table 10 cancers-16-04262-t010:** Standardized Mortality Ratios (SMRs) for Non-Cancer Causes for Cardia in Chemotherapy Patients.

Causes	<1 Year		1–2 Years		2–5 Years		>5 Years		Total		Total	
	Observed (n)	SMR (95%CI)	Observed (n)	SMR (95%CI)	Observed (n)	SMR (95%CI)	Observed (n)	SMR (95%CI)	Observed (n)	SMR (95%CI)	Patients	Mean Age at Event
Non-GC	229	20.24^p^ (17.7–23.03)	190	24.31^p^ (20.98–28.03)	208	14.95^p^ (12.99–17.13)	46	4.11^p^ (3.01–5.48)	673	15.21^p^ (14.09–16.41)	1853	70.44
GC	132	571.57^p^ (478.23–677.81)	100	634.21^p^ (516.02–771.37)	80	288.35^p^ (228.64–358.87)	16	75.16^p^ (42.96–122.06)	328	373.18^p^ (333.88–415.83)	1853	70.16
All Causes of Death	422	9.30^p^ (8.44–10.24)	332	10.54^p^ (9.43–11.73)	370	6.37^p^ (5.74–7.06)	130	2.77^p^ (2.32–3.29)	1254	6.90^p^ (6.52–7.29)	1853	71.24
Non-cancer causes	61	1.80^p^ (1.38–2.32)	42	1.78^p^ (1.29–2.41)	82	1.87^p^ (1.49–2.32)	68	1.92^p^ (1.49–2.43)	253	1.85^p^ (1.63–2.09)	1853	74.74
Septicemia	4	6.17^p^ (1.68–15.8)	1	2.21 (0.06–12.33)	0	0 (0–4.42)	0	0 (0–5.5)	5	1.92 (0.62–4.48)	1853	69.33
Other Infectious and Parasitic Diseases including HIV	1	2.45 (0.06–13.67)	4	14.25^p^ (3.88–36.5)	2	4.07 (0.49–14.68)	2	5.46 (0.66–19.72)	9	5.82^p^ (2.66–11.05)	1853	69.2
Diabetes Mellitus	0	0 (0–2.6)	0	0 (0–3.75)	1	0.55 (0.01–3.09)	2	1.35 (0.16–4.89)	3	0.53 (0.11–1.54)	1853	76.88
Alzheimer’s (ICD-9 and 10 only)	0	0 (0–2.73)	0	0 (0–3.8)	2	1.03 (0.12–3.71)	2	1.22 (0.15–4.4)	4	0.68 (0.18–1.73)	1853	86.37
Cardiovascular Diseases	22	1.71^p^ (1.07–2.59)	9	1.02 (0.46–1.93)	33	2.02^p^ (1.39–2.84)	28	2.16^p^ (1.44–3.12)	92	1.81^p^ (1.46–2.21)	1853	77.29
Cerebrovascular Diseases	3	1.3 (0.27–3.8)	0	0 (0–2.33)	1	0.34 (0.01–1.9)	2	0.85 (0.1–3.09)	6	0.66 (0.24–1.43)	1853	78.45
Pneumonia and Influenza	5	4.97^p^ (1.61–11.6)	4	5.85^p^ (1.59–14.98)	3	2.33 (0.48–6.82)	2	2.02 (0.25–7.31)	14	3.53^p^ (1.93–5.93)	1853	68.98
Chronic Obstructive Pulmonary Disease and Allied Cond	5	1.68 (0.55–3.93)	6	2.90^p^ (1.07–6.32)	10	2.65^p^ (1.27–4.88)	11	3.55^p^ (1.77–6.35)	32	2.69^p^ (1.84–3.8)	1853	76.63
Chronic Liver Disease and Cirrhosis	1	1.82 (0.05–10.16)	1	2.59 (0.07–14.44)	2	2.89 (0.35–10.43)	1	1.77 (0.04–9.84)	5	2.28 (0.74–5.32)	1853	73.35
Nephritis, Nephrotic Syndrome and Nephrosis	1	1.1 (0.03–6.11)	3	4.74 (0.98–13.87)	3	2.55 (0.53–7.46)	4	4.31^p^ (1.18–11.04)	11	3.02^p^ (1.51–5.4)	1853	73.94
Accidents and Adverse Effects	4	2.91 (0.79–7.46)	2	2.08 (0.25–7.5)	4	2.21 (0.6–5.66)	3	2.02 (0.42–5.9)	13	2.31^p^ (1.23–3.95)	1853	73.14
Suicide and Self-Inflicted Injury	1	2.42 (0.06–13.47)	2	6.95 (0.84–25.11)	0	0 (0–7.14)	1	2.41 (0.06–13.41)	4	2.45 (0.67–6.27)	1853	64.85
Other Cause of Death	14	1.84^p^ (1.01–3.09)	10	1.85 (0.89–3.41)	21	2.03^p^ (1.26–3.11)	10	1.17 (0.56–2.15)	55	1.73^p^ (1.3–2.25)	1853	72.24

^P^: means its significant (<0.05).

**Table 11 cancers-16-04262-t011:** Standardized Mortality Ratios (SMRs) for Non-Cancer Causes for Non-Cardia in Chemotherapy Patients.

Causes	<1 Year		1–2 Years		2–5 Years		>5 Years		Total		Total	
	Observed (n)	SMR (95%CI)	Observed (n)	SMR (95%CI)	Observed (n)	SMR (95%CI)	Observed (n)	SMR (95%CI)	Observed (n)	SMR (95%CI)	Patients	Mean Age at Event
Non-GC	8	1.36 (0.59–2.68)	9	2.12 (0.97–4.02)	7	0.77 (0.31–1.58)	13	1.04 (0.55–1.77)	37	1.16 (0.82–1.6)	1209	72.45
GC	153	1026.41^p^ (870.22–1202.55)	136	1268.85^p^ (1064.57–1500.91)	108	472.05^p^ (387.23–569.92)	22	72.87^p^ (45.67–110.32)	419	532.43^p^ (482.67–585.94)	1209	68.35
All Causes of Death	173	7.37^p^ (6.31–8.55)	166	9.89^p^ (8.44–11.51)	148	4.06^p^ (3.44–4.77)	80	1.40^p^ (1.11–1.74)	567	4.24^p^ (3.9–4.6)	1209	69.84
Non-cancer causes	12	0.69 (0.36–1.2)	21	1.69^p^ (1.05–2.58)	33	1.22 (0.84–1.71)	45	1.02 (0.74–1.36)	111	1.1 (0.9–1.32)	1209	74.6
Septicemia	1	2.76 (0.07–15.36)	1	3.88 (0.1–21.61)	0	0 (0–6.54)	1	1.17 (0.03–6.54)	3	1.47 (0.3–4.31)	1209	59.14
Other Infectious and Parasitic Diseases including HIV	0	0 (0–14.88)	0	0 (0–20.42)	3	8.01^p^ (1.65–23.41)	1	2.12 (0.05–11.83)	4	3.14 (0.86–8.04)	1209	68.02
Diabetes Mellitus	1	1.19 (0.03–6.66)	3	5.06^p^ (1.04–14.78)	0	0 (0–2.86)	3	1.59 (0.33–4.64)	7	1.52 (0.61–3.13)	1209	70.5
Alzheimer’s (ICD-9 and 10 only)	0	0 (0–5.34)	0	0 (0–7.56)	0	0 (0–3.29)	1	0.41 (0.01–2.28)	1	0.21 (0.01–1.17)	1209	90.67
Cardiovascular Diseases	4	0.62 (0.17–1.58)	5	1.09 (0.35–2.55)	11	1.12 (0.56–2.01)	14	0.88 (0.48–1.48)	34	0.92 (0.64–1.29)	1209	78.95
Cerebrovascular Diseases	1	0.76 (0.02–4.23)	1	1.08 (0.03–6.04)	1	0.5 (0.01–2.79)	5	1.5 (0.49–3.5)	8	1.06 (0.46–2.08)	1209	73.54
Pneumonia and Influenza	0	0 (0–6.98)	1	2.66 (0.07–14.84)	2	2.51 (0.3–9.08)	1	0.76 (0.02–4.23)	4	1.33 (0.36–3.4)	1209	68.16
Chronic Obstructive Pulmonary Disease and Allied Cond	2	1.44 (0.17–5.19)	2	1.97 (0.24–7.12)	3	1.36 (0.28–3.96)	4	1.18 (0.32–3.01)	11	1.37 (0.68–2.45)	1209	69.87
Chronic Liver Disease and Cirrhosis	0	0 (0–13.27)	0	0 (0–17.67)	3	6.71^p^ (1.38–19.62)	1	1.92 (0.05–10.71)	4	2.75 (0.75–7.04)	1209	73.62
Nephritis, Nephrotic Syndrome and Nephrosis	0	0 (0–7.19)	0	0 (0–10.26)	0	0 (0–4.69)	1	0.79 (0.02–4.42)	1	0.34 (0.01–1.91)	1209	60.34
Accidents and Adverse Effects	0	0 (0–5.11)	1	1.88 (0.05–10.48)	4	3.49 (0.95–8.93)	2	1.16 (0.14–4.2)	7	1.7 (0.68–3.5)	1209	77.75
Suicide and Self-Inflicted Injury	0	0 (0–20.91)	0	0 (0–27.66)	0	0 (0–13.07)	0	0 (0–10.9)	0	0 (0–3.96)	1209	
Other Cause of Death	3	0.77 (0.16–2.26)	7	2.51^p^ (1.01–5.17)	6	0.97 (0.35–2.1)	11	1.02 (0.51–1.82)	27	1.14 (0.75–1.66)	1209	75.35

^P^: means its significant (<0.05).

## Data Availability

The datasets generated during and/or analyzed during the current study are available from the corresponding author on reasonable request.

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
