# Peer review of "Relative Survival, Conditional Survival, and Causes of Death in Patients with Early Gastric Cancer, with a Focus on Differences Between Cardia and Non-Cardia Cancer"

_cancers, 2024, doi:10.3390/cancers16244262_

Round 1

Reviewer 1 Report

Comments and Suggestions for Authors

Dear Authors, thank you for submitting you manuscript. It is well written and it focus on a cohort of patients we do not have much data, taking into account the difficulty to diagnosis these patients during their stage 1 of the disease.

There are however some points that I want to highlight that I believe need your attention. The major problem is that you do not provide data regarding how many patients of this cohort were operated, subjected to an endoscopic resection, had chemotherapy and had radiotherapy. You focus your manuscript on analysing mortality for Stage 1 gastric cancer patients, taking into account the anatomical region of the tumours, however it seems very strange do analyse mortality and not providing data regarding the type of treatments these patients had. The treatment is extremely important to analyse cancer related mortality and even non cancer related mortality.

Please take into account the following:

Line 58 – “GC causes approximately 11,130 deaths” is this annually?

Line 93 – “LV”, should be “Lv”

Line 94 – “…also reported poor prognosis of early cardia cancer compared to non-cardia cancer although it seems to have less differentiation and worse pathologic type” – why do you use the word “although” since having less differentiation and worse pathologic type seems to be in line with worse prognosis of cardia cancers?

Line 102 – “EGC” – please state the meaning of abbreviations the first time you use them in the manuscript.

Line 149 – “The final study cohort consisted of 9,721 patients diagnosed with early GC, with 4,384 individuals exhibiting CGC and 5,337 individuals presenting NCGC.” – You should provide a figure with the layout of the whole cohort, the numbers of patients in each group and the ones excluded and why.

Line 196 – “5,337”- should not it be 3,748?

Line 198 – “A significant number of patients were married…”, I could not understand why did you include this factor as you do not discuss it later in the manuscript, why do you think being married or not, might be relevant for mortality in these patients?

Line 203 – Table 1 is probably incomplete. You do not stat which data is related to cardia and which is related to non cardia patients. You used the word “persons” which is not very commonly used, should not it be patients or individuals? It is not clear what data presented as “observed” is related to. You should also add percentages in the table.

Line 211 – Figure 1 presents a line for 120% of proportion of deaths, you should take that line off as 100% is the maximum possible. Looking at this figure I notice the majority of patients dying at 1 year died due to gastric cancer. Taking into account that these patients were stage 1, this makes me wonder if they were wrongly staged, or if they were submitted to a less efficient treatment (which you do not provide data).

Line 220 – “In the case of CGC, the most deaths (1,063; 40.4%) happened in the first year after the diagnosis of GC. Subsequently, 550 (20.9%) deaths occurred within the timeframe of 1 to 2 years, followed by 660 (25.1%) deaths within 2 to 5 years, and 357 (13.6%) deaths transpired beyond 5 years from the point of diagnosis. Among the included patients in the study, 665 (25.3%) died due to GC, whereas 1,251 (47.6%) were attributed to non-gastric cancer causes. Additionally, 714 (27.1%) were accounted for by non-cancer-related factors. Notably, the occurrence of deaths stemming from non-cancer causes exceeded those attributed to gastric cancer by a margin of 2%. 227 Within the context of NCGC, the highest number of deaths (1,102; 40.1%) occurred in the first year following the diagnosis of GC. This pattern was followed by 447 (16.2%) deaths transpiring within the interval of 1 to 2 years, 629 (22.9%) deaths transpiring within 2 to 5 years, and 573 (20.8%) deaths occurring more than 5 years after diagnosis.  Among the analysed patients, 1,490 (54.2%) died due to GC, while 242 (8.8%) were attributed to non-gastric cancer causes. Furthermore, 1,019 (37%) were attributable to non-cancer-related factors. Notably, the prevalence of deaths stemming from non-cancer causes was slightly lower than that attributed to gastric cancer.” These data should also be presented as a figure (or add these data to figure 1).

Line 232 – please remove space

Line 236 – by a margin of what? You presented this data in line 227, please present it here also.

Line 239 – “The prevailing non-gastric cancer cause was esophageal cancer…” – where these patients previously subjected to endoscopic resections? And what was the resection margin status for the operated patients?

Line 261 - You used the word “persons” which is not very commonly used, should not it be patients or individuals?

Line 326 - You used the word “persons” which is not very commonly used, should not it be patients or individuals?

Line 334 – In the supplementary file tables you used the word “persons” which is not very commonly used, should not it be patients or individuals? You should provide data regarding patients treated with surgery, chemotherapy and radiotherapy in the main manuscript and not in the supplementary file as these are extremely important for the analysis. It is also important to clarify if patients subjected to chemotherapy and radiotherapy were also operated.

Line 369 – “GC is an important cancer to study, responsible for more than one million new cases and about 769,000 deaths in 2020.” Is this globally?

Line 373 – “The study included a total of 9,721 patients diagnosed with GC, with 4,373 cases attributed to cardia cancer and 5,337 cases linked to non-cardia cancer.” This is a repetition of data and should be removed.

Line 380 – “LV” – should be “Lv”.

Line 398 – “The most common non-GC cancer cause of death in the non-cardia cancer group was miscellaneous malignant diseases.” - Where these patients previously subjected to endoscopic resections? And what was the resection margin status for the operated patients?

Line 419 – “GIT” - please state the meaning of abbreviations the first time you use them in the manuscript.

Line 462 – “CGC patients experienced a more modest increase, around 10%, during the same period. This discrepancy in the rate of increase not only points towards a potential indicator of CGC prognosis but also hints at variations in the effectiveness of treatments provided to the two groups.” – You did not provide data regarding treatments provided to the 2 groups.

Line 481 – “w”, should be “W”, capital letter.

Line 486 – “COD “- please state the meaning of abbreviations the first time you use them in the manuscript.

Line 489 – “Additionally, our analysis extends to EGC, emphasizing the critical importance of early diagnosis and surgical resection in the management of GC” – your data do not provide any information regarding early diagnosis and surgical resection, hence your conclusion should be re-evaluated.

Line 526 – Please delete this line and “1”.

Line 539 – Some references have PMID, some do not, some references provide DOI, some do not – please provide references in the same format.

Author Response

Dear Authors, thank you for submitting your manuscript. It is well written and it focus on a cohort of patients we do not have much data, taking into account the difficulty to diagnosis these patients during their stage 1 of the disease. There are however some points that I want to highlight that I believe need your attention.

The major problem is that you do not provide data regarding how many patients of this cohort were operated, subjected to an endoscopic resection, had chemotherapy and had radiotherapy. You focus your manuscript on analysing mortality for Stage 1 gastric cancer patients, taking into account the anatomical region of the tumours, however it seems very strange do analyse mortality and not providing data regarding the type of treatments these patients had. The treatment is extremely important to analyse cancer related mortality and even non cancer related mortality.

Please take into account the following:

  1. Line 58 – “GC causes approximately 11,130 deaths” is this annually?

REPLY: Thank you very much for your comment. This rate represents the deaths of 2023 only, to avoid any misunderstanding we have edited it in the manuscript.

  1. Line 93 – “LV”, should be “Lv”

REPLY: Thank you very much for your suggestion. LV et al has changed as suggested (Lv et al).

  1. Line 94 – “…also reported poor prognosis of early cardia cancer compared to non-cardia cancer although it seems to have less differentiation and worse pathologic type” – why do you use the word “although” since having less differentiation and worse pathologic type seems to be in line with worse prognosis of cardia cancers?

REPLY: Thank you very much for your comment. We agree that the use of "although" may have created an unintended contrast, as poorer differentiation and worse pathologic types indeed align with a worse prognosis for cardia cancers. We have revised the sentence to more accurately reflect the relationship between these factors and prognosis without implying a contradiction.

  1. Line 102 – “EGC” – please state the meaning of abbreviations the first time you use them in the manuscript.

REPLY: Thank you so much for your comment. EGC means early gastric cancer. To minimize the abbreviations number in the text we used early GC in the whole text.

  1. Line 149 – “The final study cohort consisted of 9,721 patients diagnosed with early GC, with 4,384 individuals exhibiting CGC and 5,337 individuals presenting NCGC.” – You should provide a figure with the layout of the whole cohort, the numbers of patients in each group and the ones excluded and why.

REPLY: Thank you very much for your suggestion. This is a SEER database related study, which is a database that include all the cancer patients in the USA, when we got the data we have selected only the patients of interest (CGC AND NCGC), so the SEERstat software excluded the others by default. You can check our previously published SEER based papers and the one published in JAMA to ensure the robustness of our methodology. 

Our papers:

  • Survival characteristics of Wilms Tumor, a reference developed from a longitudinal cohort study (https://link.springer.com/article/10.1186/s13052-024-01698-7)
  • Specific causes of death among patients with cardiac sarcoma in the United States—An analysis of The Surveillance, Epidemiology, and End Results (SEER) Program (https://onlinelibrary.wiley.com/doi/full/10.1111/jocs.16857)
  • Stroke as a cause of death in patients with cancer: a SEER-based study (https://www.sciencedirect.com/science/article/abs/pii/S1052305723001775)

Our JAMA Paper:

  • Causes of Death Among Patients With Metastatic Prostate Cancer in the US From 2000 to 2016 (https://jamanetwork.com/journals/jamanetworkopen/fullarticle/2782721\)

  1. Line 196 – “5,337”- should not it be 3,748?

REPLY: Thank you very much for your comment. Yes, it should be 3,748 and has been changed in the text.

  1. Line 198 – “A significant number of patients were married…”, I could not understand why did you include this factor as you do not discuss it later in the manuscript, why do you think being married or not, might be relevant for mortality in these patients?

REPLY: Thank you very much for your comment. Actually, according to the results of a meta-analysis published in 2023, better survival was observed in married patients when compared to unmarried (single, never‐married, divorced/separated, and widowed) in overall and cancer‐specific survival (a). That’s why we thought it would be better to add it to our manuscript since these data may be helpful for meta-analysers in the future.

  1. Krajc K, Miroševič Š, Sajovic J, et al. Marital status and survival in cancer patients: A systematic review and meta-analysis. Cancer Med. 2023;12(2):1685-1708. doi:10.1002/cam4.5003

  1. Line 203 – Table 1 is probably incomplete. You do not stat which data is related to cardia and which is related to non cardia patients. You used the word “persons” which is not very commonly used, should not it be patients or individuals? It is not clear what data presented as “observed” is related to. You should also add percentages in the table.

REPLY: Thank you very much for your comment. We have edited the table, added which is CGC and which is NCGC, also we used the word patient instead of person. The percentages also have been added.

  1. Line 211 – Figure 1 presents a line for 120% of proportion of deaths, you should take that line off as 100% is the maximum possible. Looking at this figure I notice the majority of patients dying at 1 year died due to gastric cancer. Taking into account that these patients were stage 1, this makes me wonder if they were wrongly staged, or if they were submitted to a less efficient treatment (which you do not provide data).

REPLY: Thank you very much for your comment. The figure 1 has been deleted within the recommendation of the other reviewers. A new figure has been made and added.

  1. Line 220 – “In the case of CGC, the most deaths (1,063; 40.4%) happened in the first year after the diagnosis of GC. Subsequently, 550 (20.9%) deaths occurred within the timeframe of 1 to 2 years, followed by 660 (25.1%) deaths within 2 to 5 years, and 357 (13.6%) deaths transpired beyond 5 years from the point of diagnosis. Among the included patients in the study, 665 (25.3%) died due to GC, whereas 1,251 (47.6%) were attributed to non-gastric cancer causes. Additionally, 714 (27.1%) were accounted for by non-cancer-related factors. Notably, the occurrence of deaths stemming from non-cancer causes exceeded those attributed to gastric cancer by a margin of 2%. 227 Within the context of NCGC, the highest number of deaths (1,102; 40.1%) occurred in the first year following the diagnosis of GC. This pattern was followed by 447 (16.2%) deaths transpiring within the interval of 1 to 2 years, 629 (22.9%) deaths transpiring within 2 to 5 years, and 573 (20.8%) deaths occurring more than 5 years after diagnosis.  Among the analysed patients, 1,490 (54.2%) died due to GC, while 242 (8.8%) were attributed to non-gastric cancer causes. Furthermore, 1,019 (37%) were attributable to non-cancer-related factors. Notably, the prevalence of deaths stemming from non-cancer causes was slightly lower than that attributed to gastric cancer.” These data should also be presented as a figure (or add these data to figure 1).

REPLY: Thank you very much for your comment. These data have been presented in chart bar and has been added to the manuscript.

  1. Line 232 – please remove space

REPLY: Thank you very much for your comment. The space has been removed.

  1. Line 236 – by a margin of what? You presented this data in line 227, please present it here also.

REPLY: Thank you very much for your comment. The margin has been calculated and added.

  1. Line 239 – “The prevailing non-gastric cancer cause was esophageal cancer…” – where these patients previously subjected to endoscopic resections? And what was the resection margin status for the operated patients?

REPLY: Thank you very much for your comment. We analyzed the data for various demographic and tumor characteristics, but did not include treatment details, as such information is not available in the SEER database (a,b). We have added this limitation to the Limitation section of the manuscript to ensure transparency in our analysis.

  1. Jairam V, Park HS. Strengths and limitations of large databases in lung cancer radiation oncology research. Transl Lung Cancer Res. 2019;8(Suppl 2):S172-S183. doi:10.21037/tlcr.2019.05.06
  2. Causes of Death Among Patients With Metastatic Prostate Cancer in the US From 2000 to 2016 (https://jamanetwork.com/journals/jamanetworkopen/fullarticle/2782721\)

  1. Line 261 - You used the word “persons” which is not very commonly used, should not it be patients or individuals?

REPLY: Thank you very much for your comment. This issue (observed, person) has been solved among the whole tables and now it’s (observed, Patients)

  1. Line 326 - You used the word “persons” which is not very commonly used, should not it be patients or individuals?

REPLY: Thank you very much for your comment. This issue (observed, person) has been solved among the whole tables and now it’s (observed, Patients).

  1. Line 334 – In the supplementary file tables you used the word “persons” which is not very commonly used, should not it be patients or individuals?

REPLY: Thank you very much for your comment. This issue (observed, person) has been solved among the whole tables

  1. You should provide data regarding patients treated with surgery, chemotherapy and radiotherapy in the main manuscript and not in the supplementary file as these are extremely important for the analysis. It is also important to clarify if patients subjected to chemotherapy and radiotherapy were also operated.

REPLY: Thank you very much for your comment. A paragraph Data regarding patients treated with surgery, chemotherapy and radiotherapy has been added to the manuscript. Thank you for your valuable suggestion.

  1. Line 369 – “GC is an important cancer to study, responsible for more than one million new cases and about 769,000 deaths in 2020.” Is this globally?

REPLY: Thank you very much for your comment. This data was provided from the global (36 Cancers in 185 Countries) GC study GLOBOCAN, so we can say yes its globally.

  1. Line 373 – “The study included a total of 9,721 patients diagnosed with GC, with 4,373 cases attributed to cardia cancer and 5,337 cases linked to non-cardia cancer.” This is a repetition of data and should be removed.

REPLY: Thank you very much for your suggestion. The sentence has been removed as suggested.

  1. Line 380 – “LV” – should be “Lv”.

REPLY: Thank you very much for your suggestion. LV et al has changed as suggested (Lv et al).

  1. Line 398 – “The most common non-GC cancer cause of death in the non-cardia cancer group was miscellaneous malignant diseases.” - Where these patients previously subjected to endoscopic resections? And what was the resection margin status for the operated patients?

REPLY: Thank you very much for your comment. We analyzed the data for various demographic and tumor characteristics, but did not include treatment details, as such information is not available in the SEER database (a,b). We have added this limitation to the Limitation section of the manuscript to ensure transparency in our analysis.

  1. Jairam V, Park HS. Strengths and limitations of large databases in lung cancer radiation oncology research. Transl Lung Cancer Res. 2019;8(Suppl 2):S172-S183. doi:10.21037/tlcr.2019.05.06
  2. Causes of Death Among Patients With Metastatic Prostate Cancer in the US From 2000 to 2016 (https://jamanetwork.com/journals/jamanetworkopen/fullarticle/2782721\)

  1. Line 419 – “GIT” - please state the meaning of abbreviations the first time you use them in the manuscript.

REPLY: Thank you very much for your comment.  GIT is the abbreviation of the Gastrointestinal tract, to minimize the abbreviations number in the text we used “Gastrointestinal tract” in the text.

  1. Line 462 – “CGC patients experienced a more modest increase, around 10%, during the same period. This discrepancy in the rate of increase not only points towards a potential indicator of CGC prognosis but also hints at variations in the effectiveness of treatments provided to the two groups.” – You did not provide data regarding treatments provided to the 2 groups.

REPLY: Thank you very much for your comment.  The section of this paragraph regarding the treatment has been removed.

  1. Line 481 – “w”, should be “W”, capital letter.

REPLY: Thank you very much for your comment.  The letter has been capitalized.

  1. Line 486 – “COD “- please state the meaning of abbreviations the first time you use them in the manuscript.

REPLY: Thank you very much for your comment.  COD stands for causes of death, to minimize the abbreviations number in the text we used “causes of death” in the text.

  1. Line 489 – “Additionally, our analysis extends to EGC, emphasizing the critical importance of early diagnosis and surgical resection in the management of GC” – your data do not provide any information regarding early diagnosis and surgical resection, hence your conclusion should be re-evaluated.

REPLY: Thank you very much for your comment.  The whole conclusion section has revised in accordance within our results.

  1. Line 526 – Please delete this line and “1”.

REPLY: Thank you very much for your comment. The ‘’1’’ has been removed.

  1. Line 539 – Some references have PMID, some do not, some references provide DOI, some do not – please provide references in the same format.

REPLY: Thank you very much for your comment. The references have been checked and reformatted in the same format in accordance within the guidelines of MDPI Cancers.

We hope this revision addresses your concerns and enhances the clarity and comprehensiveness of our study. Thank you again for your insightful suggestions.

Reviewer 2 Report

Comments and Suggestions for Authors

The article Relative Survival, Conditional Survival, and Causes of Death in Patients with Early Gastric Cancer, with a Focus on Differences Between Cardia and Non-Cardia Cancer presents up-to-date information on early gastric cancer. Recommendations:

1.       The introduction should be shortened to clearly present the study's background without including excessive details that would be more appropriate in the discussion section.

2.       Add a flowchart in Chapter 2.

3.       The statistical analysis is weak. Include more complex logistic regressions, ROC curves, Kaplan-Meier survival curves and calculate the power of the study to validate its relevance.

4.       The exclusion criteria are not clearly presented, potentially introducing bias in patient selection, which could invalidate the presented data.

5.       Remove Figure 1 as it provides weak information and lacks statistical significance.

6.       The discussion chapter is underdeveloped and requires clearer presentation of controversies in the literature, as well as similar studies and their findings.

7.       Discuss the implication of microRNAs as a future perspective in gastric cancer diagnosis – recommended article: 10.3390/ijms25147898

8.       Address the role of new imaging methods in improving survival rates in gastric cancer through differential diagnosis with other pathologies – recommended articles: 10.3390/diagnostics14070675

9.       Rewrite and condense the conclusions to clearly present the study’s findings.

10.   The bibliography is outdated and needs to be updated with recent references.

Author Response

The article Relative Survival, Conditional Survival, and Causes of Death in Patients with Early Gastric Cancer, with a Focus on Differences Between Cardia and Non-Cardia Cancer presents up-to-date information on early gastric cancer. Recommendations:

  1. The introduction should be shortened to clearly present the study's background without including excessive details that would be more appropriate in the discussion section.

REPLY: Thank you so much for your suggestion. We have revised the introduction to present the study's background more succinctly, focusing on the essential details and removing excessive information.

  1. 2. Add a flowchart in Chapter 2.

REPLY: Thank you so much for your suggestion. A flowchart has been created and added to the Method section.

  1. The statistical analysis is weak. Include more complex logistic regressions, ROC curves, Kaplan-Meier survival curves and calculate the power of the study to validate its relevance.

REPLY: Thank you very much for your comment. We have used the Standardized Mortality Ratio (SMR) in our analysis, as this is the most appropriate measure for analyzing the causes of death in cancer patients. The SMR allowed us to compare the observed mortality rate in our study cohort to the expected mortality rate from the general population, providing important insight into mortality patterns that are specific to the disease under investigation. Additionally, we conducted Relative Survival analysis, which is critical in understanding survival outcomes while adjusting for the background mortality rates. The Overall Relative Survival and Conditional Relative Survival (1-year and 5-year survival rates) were also calculated. This method is essential because it helps account for the effect of non-cancer-related mortality, providing a clearer picture of survival rates among cancer patients. Specifically, the Conditional Survival analysis examines the survival probabilities at various time points, adjusting for patients who have survived up to that point, which gives a more refined and clinically relevant understanding of long-term outcomes. Moreover, the SEER database offers a robust analytic framework tailored for cancer survival and mortality outcomes, and we leveraged this to conduct more accurate and context-specific analyses. The primary aim of our study remains to assess mortality causes, survival rates, and how various variables impact patient outcomes. Power and sample size calculations also are not required for SEER. given its population-based design and comprehensive data collection. As outlined in the SEER program documentation, its population-based design inherently captures all cancer cases within the covered geographic regions, ensuring complete ascertainment of data. This comprehensive data collection eliminates the need for traditional power and sample size calculations typically required for sampled populations. The robustness of SEER data stems from its routine collection of incidence, survival, and mortality information, representing approximately 26% of the U.S. population, which includes diverse demographic subgroups. This design ensures sufficient statistical power for reliable analyses without sampling biases or underrepresentation concerns. In accordance within these analyses, we aimed to address the broader question of how different factors, including tumor type, and patient characteristics, influence survival and mortality outcomes in gastric cancer. We believe this approach provides a more comprehensive and statistically valid assessment of the factors influencing long-term patient outcomes. The attached references are supporting what we have discussed and written.

  1. https://scholarworks.gsu.edu/iph_facpub/132/
  2. https://seer.cancer.gov/archive/csr/1975_2003/

  1. 4. The exclusion criteria are not clearly presented, potentially introducing bias in patient selection, which could invalidate the presented data.

Thank you very much for your comment. This is a SEER database related study, which is a database that include all the cancer patients in the USA, when we got the data we have selected only the patients of interest (CGC AND NCGC), so the SEERstat software excluded the others by default. You can check our previously published SEER based papers and one that published in JAMA to ensure the robustness of our methodology. 

Our papers:

  • Survival characteristics of Wilms Tumor, a reference developed from a longitudinal cohort study (https://link.springer.com/article/10.1186/s13052-024-01698-7)
  • Specific causes of death among patients with cardiac sarcoma in the United States—An analysis of The Surveillance, Epidemiology, and End Results (SEER) Program (https://onlinelibrary.wiley.com/doi/full/10.1111/jocs.16857)
  • Stroke as a cause of death in patients with cancer: a SEER-based study (https://www.sciencedirect.com/science/article/abs/pii/S1052305723001775)

Our JAMA Paper:

  • Causes of Death Among Patients With Metastatic Prostate Cancer in the US From 2000 to 2016 (https://jamanetwork.com/journals/jamanetworkopen/fullarticle/2782721\)

  1. 5. Remove Figure 1 as it provides weak information and lacks statistical significance.

REPLY: Thank you very much for your comment. The figure 1 has been deleted within the recommendation of the other reviewers. A new figure has been made and added.

  1. 6. The discussion chapter is underdeveloped and requires clearer presentation of controversies in the literature, as well as similar studies and their findings.

REPLY: Thank you very much for your valuable comment. We have tried our best present all the controversies and similar studies and their findings in the literature, however, this is a new area and no many literature regarding it, please let us know if any further comments or suggestions regarding the discussion. 

  1. Discuss the implication of microRNAs as a future perspective in gastric cancer diagnosis – recommended article: 10.3390/ijms25147898

REPLY: Thank you very much for your valuable suggestion. We have discussed the implication of microRNAs as a future perspective in gastric cancer diagnosis as suggested.

  1. 8. Address the role of new imaging methods in improving survival rates in gastric cancer through differential diagnosis with other pathologies – recommended articles: 10.3390/diagnostics14070675

REPLY: Thank you very much for your valuable suggestion. We have discussed the new imaging methods in improving survival rates in gastric cancer through differential diagnosis with other pathologies as suggested.

  1. Rewrite and condense the conclusions to clearly present the study’s findings.

REPLY: Thank you very much for your valuable suggestion. The whole conclusions have been rewritten as suggested.

  1. The bibliography is outdated and needs to be updated with recent references.

REPLY: Thank you very much for drawing our attention to such important point. We have updated our references and didn’t use any outdated reference only where absolutely necessary.

We hope this revision addresses your concerns and enhances the clarity and comprehensiveness of our study. Thank you again for your insightful suggestions.

Reviewer 3 Report

Comments and Suggestions for Authors

It is a great privilege to review this manuscript. This study utilizes the data from the SEER database, with a focus on the disparities between cardiac gastric cancer (CGC) and non-cardiac gastric cancer (NCGC). The data volume and analysis quantity in this article are considerable, which is conducive to the development of the current field. Nevertheless, after a detailed reading of this article, we consider that the following aspects can be enhanced:

1. The logic in the "Introduction" and "Discussion" sections of the paper is relatively weak. There seems to be a lack of strong correlation and appropriate connections between paragraphs, which makes it perplexing and fragmented when reading.

2. The article mentions that the researchers emphasized the significance of early diagnosis and surgical resection in the management of gastric cancer. However, the article fails to discuss the influence and differences of different treatment methods on CGC and NCGC, merely placing these data in the supplementary materials. It is hoped that the analysis and discussion in this regard can be supplemented.

3. Regarding gastric cancer, factors such as different invasion depths, the year of diagnosis, high- and low-incidence regions, family history of tumors, pathological types of tumors, tumor location, the longest diameter of tumors, degree of differentiation, lymph node metastasis, and distant organ metastasis are all crucial factors for survival rate and prognosis. This article is very comprehensive in analyzing the data of CGC and NCGC. However, if it is feasible to supplement the relevant analyses of more factors, this article could be even more excellent.

4.The tables and figures of the article can be appropriately beautified, and Figure 1 in the main text cannot well represent quantitative data

Author Response

It is a great privilege to review this manuscript. This study utilizes the data from the SEER database, with a focus on the disparities between cardiac gastric cancer (CGC) and non-cardiac gastric cancer (NCGC). The data volume and analysis quantity in this article are considerable, which is conducive to the development of the current field. Nevertheless, after a detailed reading of this article, we consider that the following aspects can be enhanced:

REPLY: Thank you very much for your time and effort during the review of our manuscript.

  1. The logic in the "Introduction" and "Discussion" sections of the paper is relatively weak. There seems to be a lack of strong correlation and appropriate connections between paragraphs, which makes it perplexing and fragmented when reading.

REPLY: Thank you very much for your valuable suggestion. Both sections have been revised based on your comments and those of the other reviewers. We believe they are now more coherent and readable.

  1. The article mentions that the researchers emphasized the significance of early diagnosis and surgical resection in the management of gastric cancer. However, the article fails to discuss the influence and differences of different treatment methods on CGC and NCGC, merely placing these data in the supplementary materials. It is hoped that the analysis and discussion in this regard can be supplemented.

REPLY: Thank you very much for your valuable comment. We have added the treatment to the results section based on your recommendation.

  1. Regarding gastric cancer, factors such as different invasion depths, the year of diagnosis, high- and low-incidence regions, family history of tumors, pathological types of tumors, tumor location, the longest diameter of tumors, degree of differentiation, lymph node metastasis, and distant organ metastasis are all crucial factors for survival rate and prognosis. This article is very comprehensive in analyzing the data of CGC and NCGC. However, if it is feasible to supplement the relevant analyses of more factors, this article could be even more excellent.

REPLY: Thank you very much for your valuable comment and suggestion. We have analyzed and included all the available data from the SEER database; however, the requested factors are not included in the database. To emphasize the importance of these factors, we have added a paragraph in the discussion section to highlight their relevance to survival rates and prognosis.

4.The tables and figures of the article can be appropriately beautified, and Figure 1 in the main text cannot well represent quantitative data

Thank you so much for your suggestion. The figure 1 has been deleted within the recommendation of you and the other reviewers. A new figure has been made and added.

We hope this revision addresses your concerns and enhances the clarity and comprehensiveness of our study. Thank you again for your insightful suggestions.

Round 2

Reviewer 1 Report

Comments and Suggestions for Authors

Thank you for your time reviewing the manuscript. It seems more coherent now. 

I have one more suggestion:

Line 256, 663, 664 - It should be CGC and not GCC

Author Response

Thank you for your time reviewing the manuscript. It seems more coherent now.

Reply: Than you so much for your time and effort you have put during the reviweing process of our manuscript 

I have one more suggestion:

Line 256, 663, 664 - It should be CGC and not GCC

Reply: Thank you for your suggestion. The words has been changed as suggested. 

Reviewer 2 Report

Comments and Suggestions for Authors

The authors have addressed the requested changes.

Author Response

The authors have addressed the requested changes. 
Reply: Thank you so much for your valuable suggestions and comments.